# Acceptor engineering for NIR-II dyes with high photochemical and biomedical performance

Aiyan Ji[1,5], Hongyue Lou[1,5], Chunrong Qu[1], Wanglong Lu[1], Yifan Hao[2], Jiafeng Li[1], Yuyang Wu[1,3], Tonghang Chang[1,3], Hao Chen[1,3✉] & Zhen Cheng[1,3,4✉]

It is highly important and challenging to develop donor-acceptor-donor structured small-molecule second near-infrared window (NIR-II) dyes with excellent properties such as water-solubility and chem/photostability. Here, we discovery an electron acceptor, 6,7-di(thiophen-2-yl)-[1,2,5]thiadiazolo[3,4-g]quinoxaline (TQT) with highest stability in alkaline conditions, compared with conventional NIR-II building block benzobisthiadiazole (BBT) and 6,7-diphenyl-[1,2,5] thiadiazolo[3,4-g]quinoxaline (PTQ). The sulfonated hydrophilic dye, FT-TQT, is further synthesized with 2.13-fold increased quantum yield than its counterpart FT-BBT with BBT as acceptor. FT-TQT complexed with FBS is also prepared and displays a 16-fold increase in fluorescence intensity compared to FT-TQT alone. It demonstrates real-time cerebral and tumor vessel imaging capability with μm-scale resolution. Dynamic monitoring of tumor vascular disruption after drug treatment is achieved by NIR-II fluorescent imaging. Overall, TQT is an efficient electron acceptor for designing innovative NIR-II dyes. The acceptor engineering strategy provides a promising approach to design next generation of NIR-II fluorophores which open new biomedical applications.

[1] State Key Laboratory of Drug Research, Molecular Imaging Center, Shanghai Institute of Materia Medica, Chinese Academy of Sciences, Shanghai 201203, China. [2] Shanghai Institute of Technical Physics of the Chinese Academy of Sciences, Shanghai 200083, China. [3] University of Chinese Academy of Sciences, No.19 A Yuquan Road, Beijing 100049, China. [4] Bohai rim Advanced Research Institute for Drug Discovery, Yantai, Shandong 264117, China. [5]These authors contributed equally: Aiyan Ji, Hongyue Lou. ✉email: haoc@simm.ac.cn; zcheng@simm.ac.cn

Fluorescence imaging with high spatiotemporal resolution, sensitivity, and easy accessibility has been widely studied for medical diagnostics and therapeutic applications[1–5]. However, unfavorable light attenuation and auto-fluorescence in tissue at conventional visible and near infared (NIR) fluorescence imaging windows (400–900 nm) have severely compromised imaging depth and sensitivity[6,7]. Recently, owing to reduced photon scattering, low photo-absorption, and low auto-fluorescence of tissue, fluorescence imaging in the second near-infrared window (NIR-II, 1000–1700 nm) has attracted immense attention[8–13]. NIR-II imaging allows centimeters imaging depth at a micron-scale resolution of anatomic features, showing high potential to visualize deep anatomical features in vivo with high resolution and sensitivity[14].

Widely used inorganic NIR-II fluorophores such as carbon nanotubes[15,16], quantum dots[17,18], and lanthanide nano-particles[19–22] hold unknown long-term toxicity concerns that do not favor clinic translation. By contrast, organic small molecule dyes benefit from minimal toxicity and easy excretion from the body, drawing significant interests across life and medical sciences[23,24]. For organic small molecule dyes, pursuing long absorption/emission wavelength, high chem/photostability, and high quantum yield are a subject of major interest[25]. Among the organic NIR-II probes, donor-acceptor-donor (D-A-D) small molecules have been widely explored because of their highly tunable electronic structures and good optical properties, producing low-bandgap fluorophores that emit long-wavelength photons. The fluorophores can be tuned to absorption at different spectrum wavelengths by the combination of varying strength of D/A units[26–28]. In the D-A-D skeleton, the highest occupied molecular orbital (HOMO) is delocalized along the whole molecular backbone, and the lowest unoccupied molecular orbital (LUMO) is almost located on the acceptor core. Previous studies have focused on donor engineering for developing NIR-II D-A-D dyes with good performance. Various fluorophores have been designed and prepared with the introduction of shielding units and donors' steric hindrance effect[29,30]. However, acceptor moieties actually have a much more significant impact on the performance of the D-A-D fluorophores. They have been coupled with donor moieties to adjust the excited-state properties and the fluorescence production efficiency[31,32]. By applying different strategies such as using heavy atom effect[33], adding electron-withdrawing substituents[34], coupling with thienothiophene[35], and introducing fluorine atom[36], the acceptors' electron affinity has been changed.

Among limited acceptor units reported, benzobisthiadiazole (BBT) (Fig. 1a) is a widely used building block for its high electron affinity and planar configuration. The BBTD-type unit possesses a substantial quinoidal character within a conjugated backbone, allowing greater electron delocalization and thus lowering the band gap[26]. However, referring to the reported studies, most of the BBT-based polymers have relatively high HOMO energy levels making them unstable in air[37–39]. In addition, BBT has been proved to be sensitive to harsh synthetic procedures. It is labile and is decomposed in basic or reducing environment, restricting its use for the development of more advanced NIR-II fluorophores[23,40,41]. 6,7-diphenyl-[1,2,5] thiadiazolo [3,4-g] quinoxaline (PTQ) (Fig. 1a) is another acceptor widely used in medicinal chemistry and photoelectric fields[42–45]. Unfortunately, dyes with PTQ acceptor are hypsochromic shift compared with BBT dyes, though the longer-wavelength NIR light penetrates much deeper tissue[6,12]. Moreover, dyes with PTQ acceptor are all hydrophobic and need to be encapsulated in a polymer matrix which have slow in vivo clearance and does not favor their potential clinical translation as well[42–45]. Overall, acceptors with more properties such as high stability and bathochromic shift

properties are highly desirable for developing next generation NIR-II dyes. Unfortunately, little efforts has been spent on investigation of new acceptor structures.

Herein, we have explored acceptor engineering for optimizing NIR-II fluorophores and revealed the relationship between the acceptor structures and fluorescence properties. In this context, we study the photochemical properties of three electron acceptors, BBT, PTQ, and the acceptor designed in this study, 6,7-di(thiophen-2-yl)- [1,2,5] thiadiazolo [3,4-g] quinoxaline (TQT) (Fig. 1b). Among them, TQT has shown the highest alkali stability than BBT and PTQ, and it thus has been further used to couple with the donor substituted triphenylamine (TPA) to construct the NIR-II fluorophore TPA-TQT. In comparison to its counterpart CH-4T using BBT as an acceptor, which is a widely used NIR-II dye, TPA-TQT showed ultra-high stability in the presence of reactive oxygen/nitrogen species (ROS/RNS), metal ions and active biomolecules, and various alkali conditions. Inspired by the above results, we further explore the relationship between the acceptor structures and fluorescence properties. TQT showed a longer emission wavelength than PTQ. Then we theoretically calculate the bandgaps of three dyes (FT-TQT, FT-PTQ and FT-BBT) with 9,9′-dialkyl substituted fluorene as shielding unit and thiophene as the second donor (FT), which can increase the conjugation length and red-shift the fluorescence emission. The calculated bandgap of FT-TQT ($\Delta E = \sim 1.3\,\mathrm{eV}$) is similar to FT-BBT but much narrower than FT-PTQ ($\Delta E = \sim 1.6\,\mathrm{eV}$), which imply FT-TQT hold a longer emission wavelength than FT-PTQ. What's more, FT-TQT (0.49%) shows a higher quantum yield than FT-BBT (0.23%) in methanol. So, FT-TQT is subsequently selected for further research. A loading strategy with biocompatible fetal bovine serum (FBS) to the FT-TQT is employed to improve the brightness further. After loading FT-TQT into FBS, FT-TQT@FBS is demonstrated a 16-fold increase in fluorescence intensity and an 8-fold increase in quantum yield compared with FT-TQT in PBS ($\mathrm{QY_{FT\text{-}TQT@FBS}} = 0.2\%$, $\mathrm{QY_{FT\text{-}TQT}} = 0.025\%$). High-resolution imaging of cerebral vasculature and tumor vessels is achieved. Further work identified the efficacy and accuracy of identifying and real-time monitoring tumor vascular disruption after treatment with combretastatin A4 phosphate (CA4P) for the first time. Overall, this work provides a new strategy to develop advanced NIR-II fluorophores for multi-functional biological imaging.

## Results
**The comparison of different acceptor NIR-II dyes**. Most reported NIR-II D-A-D water-soluble fluorophores, including CH1055-PEG, CQ-4T, IR-E1, IR-FP8P, and TPA-T-TQ NPs, were based on BBT and PTQ as electron acceptors (Supplementary Table 2). Our study selected three electron acceptors (BBT, PTQ, and TQT) and investigated their optical properties first (Fig. 2a). By examining the absorption spectra of BBT-2Br, TQT-2Br, and PTQ-2Br in dichloromethane, we found acceptors exhibit substantial redshift of the absorption, which could be attributed to the raising of HOMO energy levels (Fig. 2b). However, the emission spectra are slightly red-shifted from PTQ-2Br, TQT-2Br to BBT-2Br (Fig. 2c).

Furtherly, the stability of BBT-2Br, TQT-2Br, and PTQ-2Br was estimated in acid-base conditions (Fig. 2d, e, Supplementary Fig. 18). BBT-2Br, TQT −2Br, and PTQ-2Br were stable in acid conditions. Nevertheless, upon adding triethylamine (TEA) to the BBT-2Br solution, the solution's appearance changed from pink to yellow immediately. Meanwhile, a new peak appeared in the high-performance liquid chromatography (HPLC) owing to the decomposition of BBT-2Br under alkaline conditions (Fig. 2e).

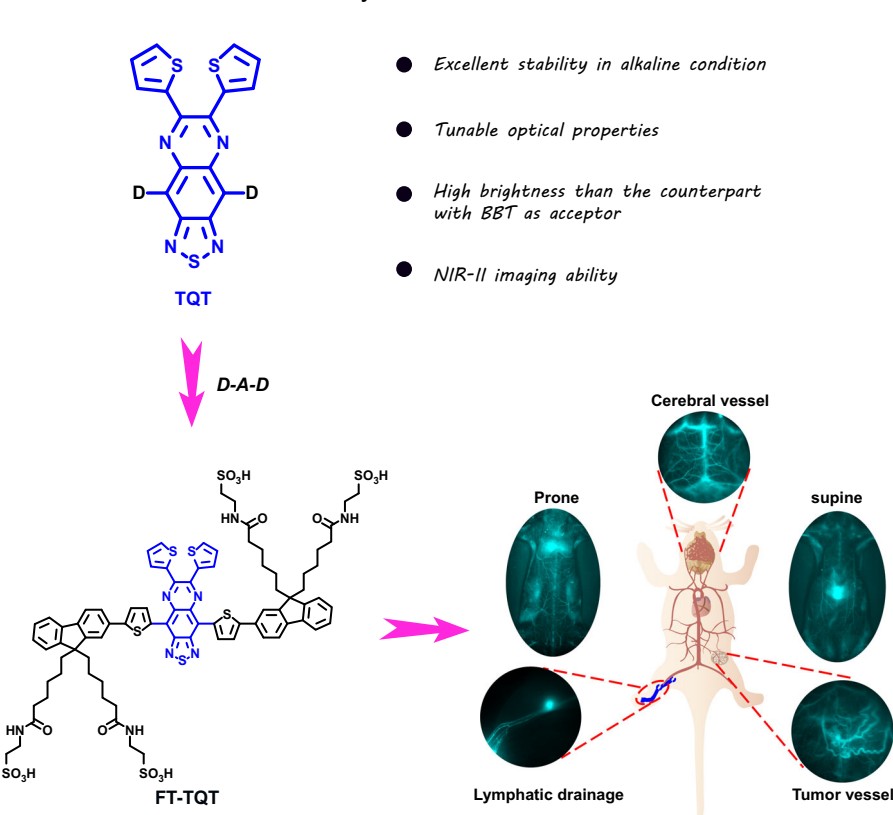

**a** Previously reported D-A-D NIR-II dyes used BBT and PTQ as electron acceptors

BBT | or | PTQ | CH-4T

**b** This work: TQT based D-A-D NIR-II dyes

TQT

- Excellent stability in alkaline condition
- Tunable optical properties
- High brightness than the counterpart with BBT as acceptor
- NIR-II imaging ability

*D-A-D*

FT-TQT

Cerebral vessel
Prone
supine
Lymphatic drainage
Tumor vessels

**Fig. 1 Designing a new generation of NIR-II D-A-D dyes by acceptor engineering strategy. a** Previous NIR-II D-A-D dyes used BBT and PTQ as electron acceptors; **b** D-A-D NIR-II fluorophore structures of this work based on TQT acceptor.

In addition, with the increase in alkali concentration, the PTQ-2Br's peak areas also decreased slightly (Supplementary Fig. 18). On the contrary, there is nearly no change in the solution appearance and HPLC spectrum of the TQT-2Br under alkaline conditions (Fig. 2d). Overall, the results showed that TQT-2Br were most stable to the basic synthetic environment compared with BBT-2Br and PTQ-2Br. Considering the photochemical properties, TQT-2Br is the desired acceptor, which shows appropriate redshift and good stabilities. To further explore the stability of the complete D-A-D fluorophores with different electron acceptors, TPA-TQT was synthesized (Fig. 2f, Supplementary Fig. 1, Supplementary Figs. 28–32). The absorption and emission spectra of TPA-TQT and CH-4T (with BBT as acceptor) in water are shown in Fig. 2g. Compared with CH-4T, TPA-TQT has a significant blue shift, which is unfavorable for NIR-II imaging. Both TPA-TQT and CH-4T exhibit higher photostability under continuous 808 nm laser irradiation for 1 h (Fig. 2h). The absorption spectra and photographic solutions of TPA-TQT and CH-4T before and after PBS treatment with different

pH (7.4, 8.0, and 8.5) values are depicted in Fig. 2i–k, respectively. After irradiation of 808 nm laser, the maximal absorption intensity of CH-4T in PBS (pH 8.5) drops about 83% relative to the original value. In contrast, there is nearly no change in the absorption spectra and solution appearance of the TPA-TQT under 808 nm laser irradiation (Fig. 2l). These results indicate that TPA-TQT is highly resistant to the basic environment, which is ideal for stable and accurate in vivo imaging (e.g., pancreas (pH 8.35–8.45), large intestine (pH 8.4–8.55)).

To explore the relationship between the acceptor structures and fluorescence properties, first, we applied theoretical calculations to the BBT, PTQ, and TQT structured D-A-D fluorophores (FTs) with the same 9,9'-dialkyl substituted fluorene as a donor (Fig. 3a). The result revealed that FT-TQT and FT-BBT had similar band gaps ($\Delta E = \sim 1.3$ eV) which is much narrower than FT-PTQ ($\Delta E = \sim 1.6$ eV). It can be attributed to TQT and BBT's stronger electron-withdrawing abilities (Fig. 3b, Supplementary Fig. 3). According to the results of Fig. 2c and theoretical calculations, so we synthesized the BBT and TQT fluorophores.

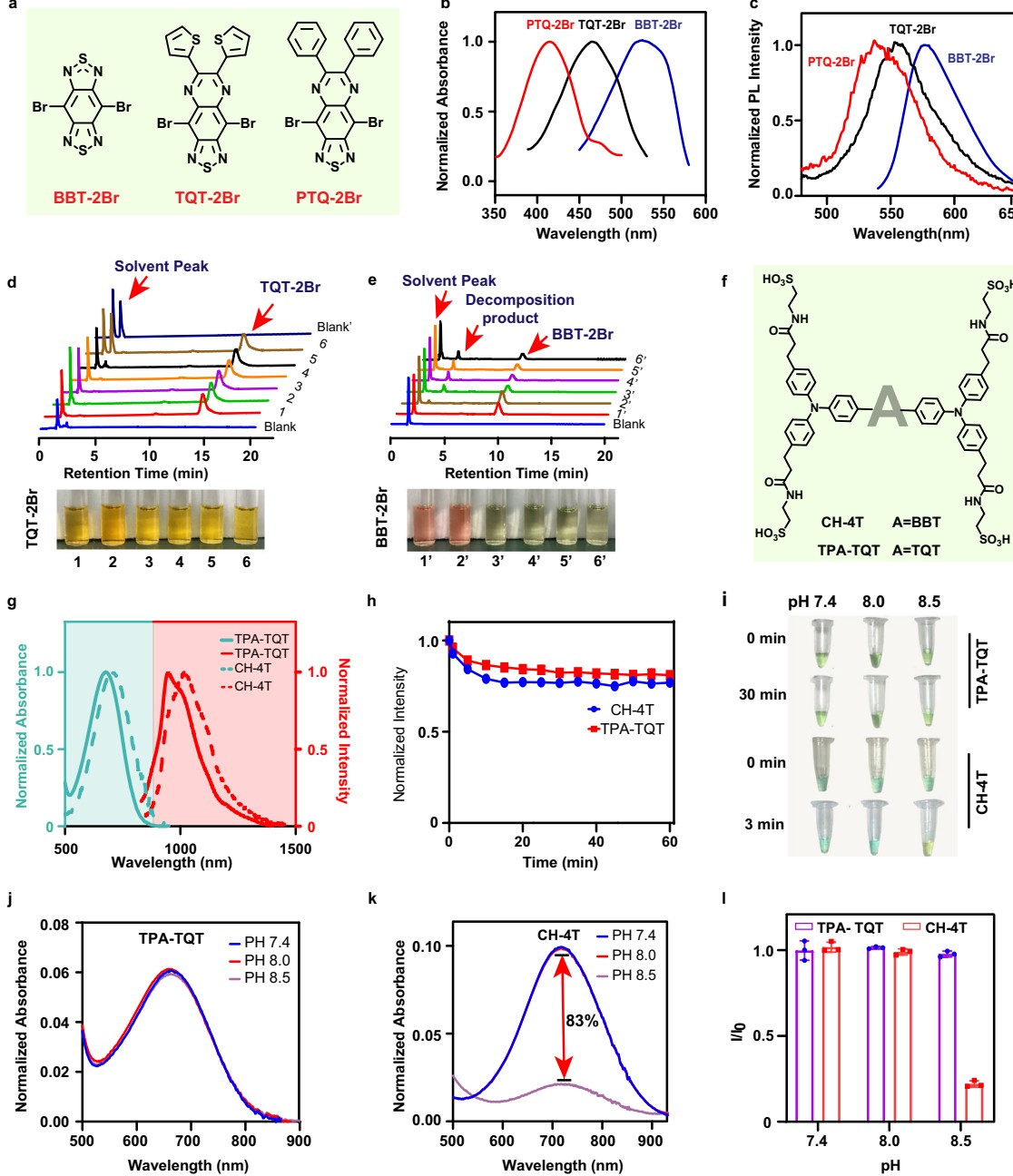

**Fig. 2 The comparison of different acceptor NIR-II dyes. a** Chemical structures of BBT-2Br, TQT-2Br, and PTQ-2Br. Absorption (**b**) and emission (**c**) spectra of BBT-2Br, TQT-2Br, and PTQ-2Br in dichloromethane. HPLC chromatograms of TQT-2Br (**d**) and BBT-2Br (**e**) at various acid-base conditions. Bright-field images of TQT-2Br and BBT-2Br (**d**, **e**) in MeOH (5% DMF, 1 mL) at various acid-base conditions. 1,1': control solution; 2,2': control solution + excess trifluoroacetic acid (TFA); 3,3': control solution + 1 μL Triethylamine (TEA); 4,4': control solution + 5 μL TEA; 5,5': control solution + 10 μL TEA; 6,6': control solution + 20 μL TEA; Blank': 50 μL DMF + 20 μL TEA + 930 μL MeOH. **f** Chemical structures of TPA-TQT and CH-4T. **g** The absorption spectra and emission spectra of TPA-TQT and CH-4T in H$_2$O. **h** Photostability of TPA-TQT and CH-4T under continuous laser irradiation (808 nm, 150 mW cm$^{-2}$). **i** Photographs of the TPA-TQT and CH-4T in PBS solutions at various pH values after 808 nm light irradiation. The absorption spectra of **j** TPA-TQT and **k** CH-4T in PBS at various pH values under 808 nm laser irradiation (150 mW cm$^{-2}$). **l** Plot of I/I$_0$ versus pH. I is the maximal NIR absorption intensity of TPA-TQT/CH-4T in PBS (pH 8.0, 8.5) solution, I$_0$ is the maximal NIR absorption intensity of TPA-TQT/CH-4T in PBS (pH 7.4) solution, respectively. Data are presented as mean ± s.d. derived from $n = 3$ independent experiments.

Key steps used to assemble the core structures of the target compound included Suzuki cross-coupling reaction, zinc reduction, and ring closure. To enhance the water solubility of the FT dyes, four sulfonic groups were introduced into the chemical structures. The detailed synthetic procedures and characterization are described in the Supporting Information (Supplementary Fig. 2 and Supplementary Figs. 33–44).

**Chemical stability evaluation of BBT and TQT based dyes.** To examine the chemical stability of BBT and TQT based dyes in the presence of reactive oxygen/nitrogen species (ROS/RNS), metal ions, and active biomolecules, we measured the absorption spectra of CH-4T, FT-BBT, TPA-TQT, and FT-TQT after incubated with the above substances at 37 °C for 1 h (Supplementary Figs. 19–21). The results showed that all dyes are stable in the

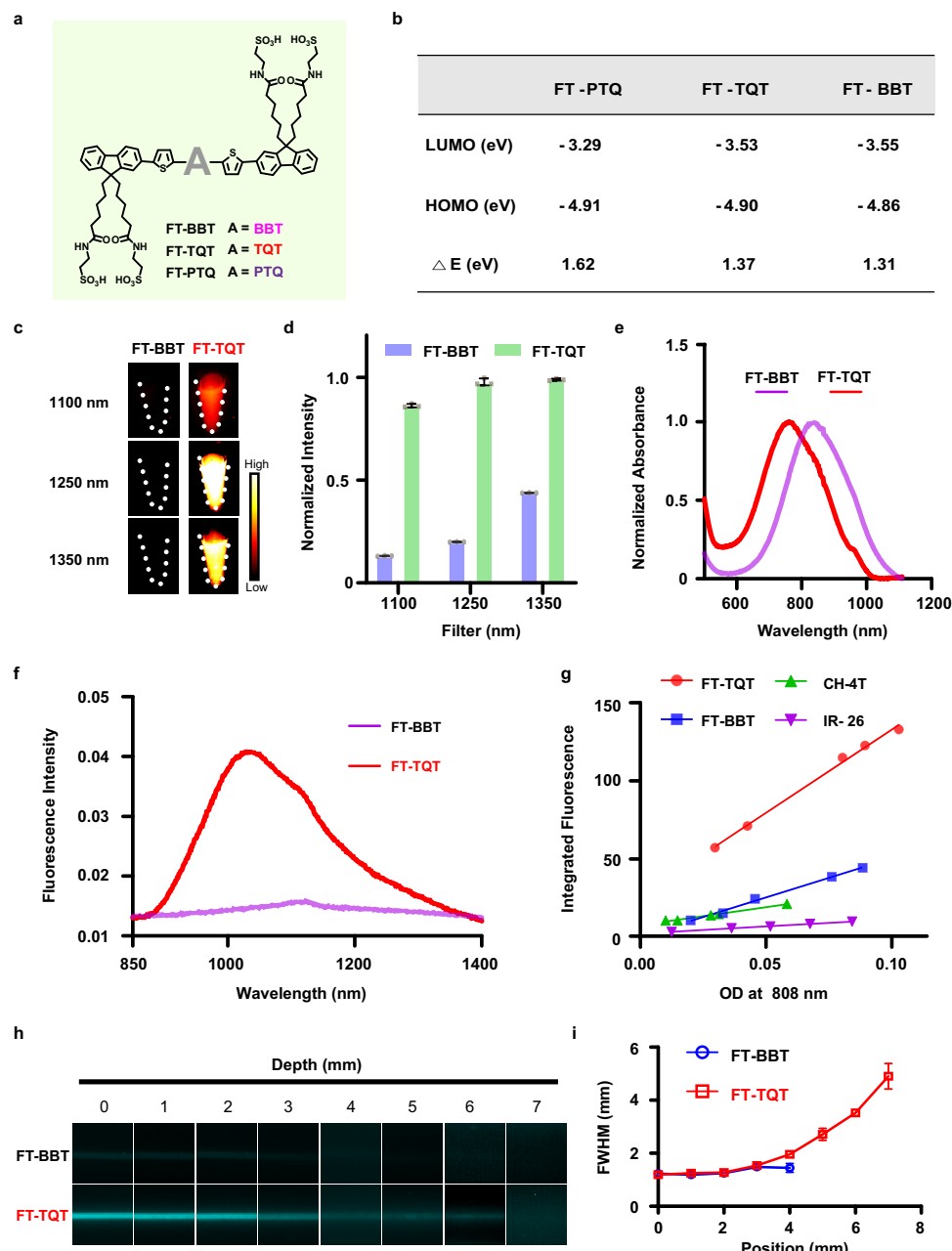

**Fig. 3 Optical characterization of NIR-II FTs. a** The structure of FT-BBT, FT-TQT, and FT-PTQ. **b** Theoretical calculations for the investigation of photophysical properties. **c** NIR-II fluorescent images of FT-BBT and FT-TQT in deionized water at equivalent absorbance of OD 0.1 at 808 nm with 1100, 1250, and 1350 nm long-pass (LP) filters. **d** Normalized fluorescent intensity of FT-BBT and FT-TQT in **c** with different filters. Data are presented as mean ± s.d. derived from $n = 3$ independent detections. **e** Normalized Absorption spectrum of FT-BBT and FT-TQT in deionized water. **f** The fluorescence emission spectrum of FT-BBT and FT-TQT in deionized water at an equivalent concentration of 10 μM. **g** Plot of the integrated fluorescence spectrum of FTs and CH-4T at five different concentrations in methanol due to low quantum yield in water. Linear fits were used to calculate quantum yield by comparing the slopes to reference IR-26 (QY = 0.05%). **h** Fluorescence images of capillaries filled with FT-BBT and FT-TQT in PBS (pH 7.4), respectively, immersed in 1% Intralipid with varying depth. Imaging signals were collected in the 1300 nm region under 808 nm excitation. **i** Wavelength-dependent full-width at half-maximum (FWHM) of cross-sectional profiles in capillary images as a function of depth. The bars represent mean ± s.d. derived from $n = 3$ independent measurements. The detailed imaging parameters for each image are listed in Supplementary Table 3.

presence of metal ions (K⁺, Na⁺, Ca²⁺, Mg²⁺, Fe²⁺, and Zn²⁺) and active biomolecules (GSH, Cys, Hcy, ascorbic acid (AA), and dehydroascorbic acid (DHA)), although CH-4T is red-shifted in the presence of Ga²⁺ and Fe²⁺. However, BBT-based dyes have shown poor stability than TQT-based dyes in the presence of ROS/RNS, especially ClO⁻. To evaluate the alkali stability of the four fluorophores, firstly, the absorption spectra of four fluorophores were tested at different time points in different alkali

solutions (Supplementary Fig. 23). CH-4T showed the worst stability in different alkali solutions, including 1%NaOH, 1%TEA, and 1%DIEA. It was degraded by 98%, 76% and 70% in 1% NaOH, 1% TEA, and 1% DIEA on day seven, respectively. Another BBT-based dye, FT-BBT showed higher stability in alkali solutions than CH-4T, which illustrates the electron donor unit plays a role in the stability of NIR-II dyes. However, FT-BBT still partially decomposed in 1%NaOH (28% on the seventh day).

Undoubtedly, TQT-based NIR-II dyes, TPA-TQT, and FT-TQT showed the best stability in all tested alkali solutions. In addition, we evaluated the changes of absorption curves of FT-BBT, TPA-TQT, and FT-TQT in NaOH solutions with different mass concentrations (Supplementary Fig. 24). In 5% NaOH solution, FT-BBT degraded 42% after incubated for 24 h. Excitedly, TQT-based dyes remain stable even if in 5%NaOH, which shows TQT-based D-A-D dyes can be widely used for chemical modification under different alkali conditions. CH-4T, FT-BBT, TPA-TQT, and FT-TQT were incubated at pH 5.0–10.0 with 37 °C water baths (Supplementary Fig. 25). Nearly half of CH-4T decomposed when incubated under pH 8.5 for 24 h. However, TPA-TQT and FT-TQT remained stable after incubated at pH 8.0 and pH 8.5 for 96 h (Supplementary Fig. 26). Besides, TPA-TQT and FT-TQT showed ultra-high photochemical stability in methanol and mouse serum (Supplementary Figs. 22, 27). However, the fluorescence intensity of FT-TQT was 9.6 times higher than that of TPA-TQT in $H_2O$ at the same concentration (Supplementary Fig. 22e). Meanwhile, the fluorescence intensity of FT-TQT increased 8.2 times after incubated with mouse serum at 37 °C for 1 h. Thus, considering the emission wavelength and stability, we chose FT-TQT and FT-BBT as the next research object to compare the differences of optical properties between BBT- and TQT-based dyes.

**Photophysical characterization of NIR-II FTs**. The NIR-II fluorescent images in Fig. 3c qualitatively show the disparate brightness levels between FT-BBT and FT-TQT, all with matching absorbance at 808 nm (OD 0.1). Quantitatively, FT-TQT showed a 6.6-, 4.9-, and 2.3- times increase in fluorescence intensity than FT-BBT at 1100 nm, 1250 nm, and 1350 nm LP filters, respectively (Fig. 3d). The absorbance spectrums of FT-BBT and FT-TQT were shown in Fig. 3e, which revealed the excitation peaks of FT-BBT and FT-TQT were ~845 nm and ~770 nm. Meanwhile, FT-TQT showed an emission peaked at 1034 nm with a tail extending into the NIR-IIa region (1300-1400 nm, Fig. 3f). However, under the same concentration (10 μM) and conditions (808 nm), FT-BBT hardly emitted fluorescence. The quantum yields of FT-TQT, FT-BBT and current NIR-II dye CH-4T were determined to be 0.49%, 0.23% and 0.11% in methanol. The solvent was chosen methanol other than water because the quantum yield of FT-BBT in water cannot be detected (with IR-26 as a reference, QY = 0.05%, Fig. 3g, Supplementary Table 1, Supplementary Figs. 5, 6). Furthermore, the superior photostability of FT-BBT and FT-TQT was observed by exposing ICG and FT-BBT and FT-TQT to the continuous laser irradiation in deionized water for 1 h, respectively (Supplementary Fig. 4). When capillary tubes filled with FT-BBT and FT-TQT solution were immersed in 1% intralipid solution at increased phantom depth under different filters (1300 nm, 1500 nm), bioimaging results of FT-TQT resolve sharper edges of the capillary at a depth up to 7 mm than that of FT-BBT (Fig. 3h, Supplementary Fig. 9). With increased penetration depth, attenuation of image intensities and blurring of capillary profiles is observed for all fluorophores. In addition, cross-sectional profiles of capillary show apparent feature integrity for FT-TQT compared with FT-BBT, which can be attributed to the higher brightness of FT-TQT (Fig. 3i, Supplementary Fig. 9).

**In vivo circulatory system and lymphatic drainage NIR-II imaging**. After estimating the optical properties of FT-BBT and FT-TQT, we turned to NIR-II in vivo investigation for further screening. High-magnification hindlimb blood vessel networks (Fig. 4a, b, Supplementary Figs. 10, 11) can be discriminated at the 1400 nm sub-NIR-II window. Not surprisingly, FT-TQT generated higher contrast for vessel imaging compared to the

FT-BBT. Collectively, the performances of FT-TQT in vitro and in vivo encouraged us to explore its imaging potentials in the following experiments.

The lymph node drainage plays a vital role in tumor metastasis. To investigate NIR-II biomedical imaging's efficacy in lymphatic drainage in vivo, FT-TQT was injected intradermally at the normal nude mice's footpad (Fig. 4c, e). Upon injecting, crowded collateral lymph vessels are observed unambiguously. Four lymph vessels with diameters from 226 to 322 μm were visualized clearly (Fig. 4d, e). In addition, the popliteal lymph node was easily identified along with its afferent lymphatic vessels. The results demonstrate FT-TQT has excellent potential for lymphatic imaging.

Visualizing accurate anatomical information of vascular structure is the premise of real-time tracking of the blood circulation system, which would help understand its dysfunction. NIR-II vascular imaging was performed by intravenous injection of FT-TQT (200 μL, 1 mg mL$^{-1}$) into the Balb/c mouse (Fig. 4f, h, Supplementary Fig. 12). The whole angiography was visualized at sequential long-pass filters from 1000 to 1400 nm with a gradually increased exposure time. The vessels at a longer wavelength are more apparent than those at a shorter wavelength due to the lower tissue absorption, scattering, and autofluorescence at the longer wavelength (Fig. 4f, g). In addition, FT-TQT was found to have a pretty long blood half-life time (~10 h), which may cause by the interaction between FT-TQT and serum proteins, such as albumin[46]. The fluorescent signals of FT-TQT in the blood can still be detected at 10 h post-injection over the 1400 nm sub-window (Fig. 4h, i, Supplementary Fig. 17). Such a long blood retention time could be used for long-term accurate blood vessel monitoring and benefit the fluorophore's accumulation in targeted tissues. Of note, fluorescence signals could be mainly observed in the liver, which provides the possibility of diagnosing liver diseases[47]. Thanks to the high brightness and long circulation properties of FT-TQT, whole-body vessel imaging was successfully conducted with high resolution, inspiring us to assess vascular-related disorders further.

**Biocompatibility**. To explore the toxicity of FT-TQT, the mouse embryonic fibroblast cell line 3T3 and human osteosarcoma cell line 143B were evaluated by standard MTT analysis in vitro. After incubation with FT-TQT for 24 h, no apparent cytotoxicity was observed in both cell lines even at high concentrations up to 100 μM, indicating its low cytotoxicity and excellent biocompatibility in vitro (Supplementary Fig. 13). Next, we evaluated in vivo toxicity by injecting FT-TQT (100 μL, 1 mg mL$^{-1}$) into normal Balb/c mice. After administration, body weight and serum biochemistry were monitored at 7 and 14 days. We observed no noticeable differences in body weight or blood markers between the FT-TQT treated and control groups (Supplementary Fig. 14). NIR-II imaging of the vital organs was performed to evaluate the biodistribution of FT-TQT. The results revealed that the dye primarily accumulates in the kidney, liver, and spleen at 7 and 14 days after intravenous administration (Supplementary Fig. 15). The H&E staining of the major organs indicated no noticeable pathological change after FT-TQT treatment (Supplementary Fig. 15). Generally, the biocompatibility of FT-TQT is excellent, showing a promising future for clinical translation.

**High-resolution NIR-II imaging for vascular network of brain and tumor**. Cerebral microvasculature imaging is a practical approach to understand cerebrovascular diseases, such as traumatic brain injury, stroke, and vascular dementia[48–50]. Fluorophore with a high quantum yield in the NIR-II window holds great promise to visualize cerebral vasculature, which is located

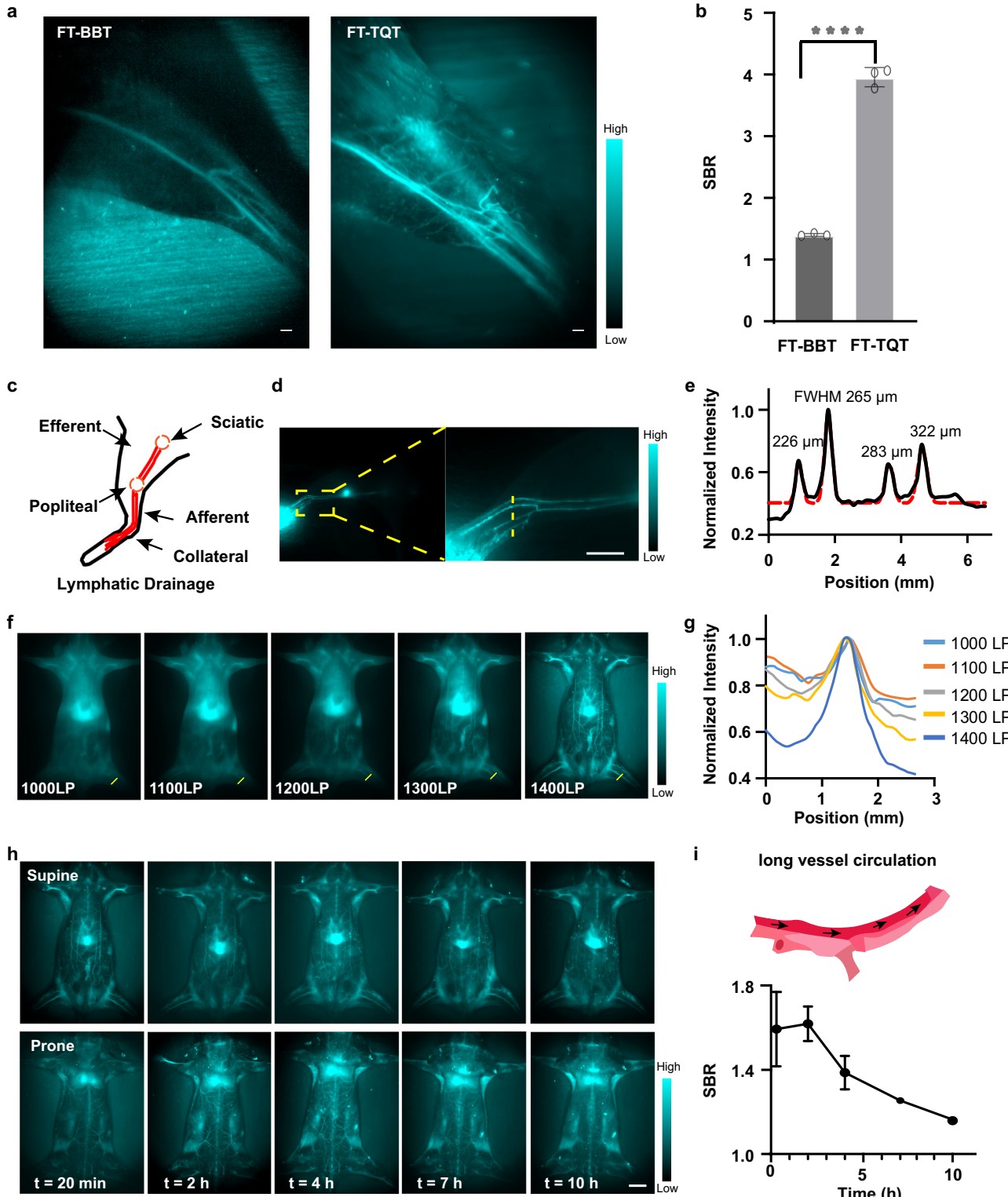

**Fig. 4 In vivo NIR-II imaging for evaluation of the circulatory system and lymphatic drainage. a** NIR-II bioimaging of mice hindlimb at 1400 nm LP filter by FTs administration. Scale bar, 1 mm. **b** SBR (vessel-to-muscle signal ratio) in balb/c mice hindlimb images by FTs administration. ****$P < 0.0001$. $P$ value was obtained from unpaired two-tailed $t$ tests. The bars represent mean ± s.d. derived from $n = 3$ independent mice. **c** Schematic illustration of the anatomical structure of the lymphatic system in the hindlimb of Balb/c mice. **d** Fluorescence images of lymphatic drainage using FT-TQT as contrast agents in the hindlimb of Balb/c mice on an InGaAs camera. Scale bar, 5 mm. **e** Cross-sectional fluorescence intensity profiles (black solid) and Gaussian fit (red dotted) along the yellow line in **d**. **f** The high brightness of the FT-TQT affords whole-body imaging with sequential LP filters from 1000 to 1400 nm. The longer NIR-II window increases the imaging quality with reduced scattering and autofluorescence. **g** Cross-sectional profiles of the vessel to normal tissue ratio across the dashed yellow lines ($70 \ \mathrm{mW \ cm^{-2}}$). **h** In vivo circulation of FT-TQT. (imaging condition: 200 s exposure time, 1400 nm window) Scale bar, 5 mm. **i** Scheme of long vessel circulation of the FT-TQT and measured the vessel to normal tissue ratio over post-injection time points. Data are presented as mean ± s.d. derived from $n = 3$ independent mice. The detailed imaging parameters for each image are listed in Supplementary Table 3.

much deeper (≈1.3 mm) relative to the skin surface[9]. It is reported that NIR-II dye with sulfonic acid functional groups readily forms supramolecular assemblies with plasma proteins to produce a brilliant increase in fluorescent brightness[10,51]. Interestingly, fluorescent signals of FT-TQT only increased obviously in the presence of fetal bovine serum (FBS) (Supplementary Fig. 7). The relative fluorescence brightness of FT-TQT/FBS heated to 70 °C for 10 min termed FT-TQT@FBS Heated, was 16-fold, 8-fold, and 11-fold brighter than FT-TQT/PBS, FT-TQT/FBS without heated, and FT-TQT/HSA, respectively (The preparation procedures and characterization of FT-TQT@FBS are described in the Supplementary Fig. 8). FT-TQT was further demonstrated to have little interaction with mouse whole blood, and red blood cells, owing to their NIR-II signals showing almost no change than FT-TQT/PBS. So, we used FT-TQT@FBS to image cerebral vasculature. As a result, the superior resolution of tiny vessels was visualized sharply through the intact scalp and skull (Fig. 5b). To obtain the detailed anatomical information, the FWHM values of the brain vessels were calculated to be 117 μm and 119 μm (1300 nm LP) (Fig. 5c). These data suggest that FT-TQT@FBS have significant advantages for NIR-II blood vessel imaging in deep tissue. To evaluate the biocompatibility of FT-TQT@FBS, NIR-II imaging of the vital organs was performed after intravenous administration at 7 and 14 days (Supplementary Fig. 16). Compared with FT-TQT, FT-TQT@FBS is most widely distributed in the liver, which has a large particle size and is hard to excrete from the kidney. However, FT-TQT is mainly distributed in the kidney, followed by the liver. In addition, the H&E staining of the major organs indicated no noticeable pathological change after FT-TQT@FBS treatment.

Tumor growth and metastasis are closely related to angiogenesis[52]. To visualize the vascular network in nude mice with xenograft osteosarcoma, we subsequently performed high-contrast, real-time NIR-II imaging using FT-TQT@FBS. The recorded NIR-II video of tumor vessels in mice produced super-contrast vessel imaging over the 1100 nm sub–NIR-II window (Fig. 5f, Supplementary Movie 1). First, the tumor's main vessel was easily identified along with its afferent irregular vascular branches, showing typical characteristics of tumor blood vessels. The tumor's main vessels' fluorescence intensity gradually attenuated with the extended time (Fig. 5g). Then, the FWHM values of the vessels (yellow dashed line) for different filters were calculated to be 114 μm (1100 nm LP), 102 μm (1350 nm LP), and 233 μm (1100 nm LP) (Fig. 5d, e, h). Finally, the vascular density of tumors was assessed at different time points using a vascular quantification algorithm based on a modified Hessian matrix method[36,53]. The density of tumor blood vessels increased gradually following the abundant branches were lighted after injected FT-TQT@FBS (Fig. 5i). These data collectively indicated that FT-TQT@FBS afforded impressive blood vessel resolution and inspired us to broaden vascular-related therapy assessment.

**Real-time monitoring of tumor vascular disruption.** Blood vessels deliver oxygen and nutrients to every part of the body but also nourish cancer. Cutting off blood supply selectively and starving the tumor has been proved to be an effective treatment strategy[54]. Tumor vascular disrupting agent (VDA), combretastatin A4 phosphate (CA4P), rapidly causes tumor vascular shutdown and subsequently triggers a cascade of tumor cell death (Fig. 6a)[55]. To monitor the tumor vascular disruption after administrating CA4P, we performed xenograft osteosarcoma blood vessel real-time NIR-II imaging using FT-TQT@FBS on nude mice. In the control group (without CA4P treatment), irregular tumor vessels were visible after 5 min PI of FT-TQT@FBS complexes (Fig. 6b). Not surprisingly, there was no

significant change in vascular morphology of the tumor until 35 minutes. In the treated group, after 30 min administration of CA4P (10 mg kg⁻¹), FT-TQT@FBS were injected and the tumor vessel's real-time images were acquired with high fidelity (Fig. 6c, Supplementary Movie 2). The shape of vasculature gradually blurred until it disappeared with time. Furthermore, the cross-sectional profiles of the tumor blood vessels at the same position were measured (Fig. 6d). Over time, two tumor vessels could not be discerned owing to the effect of CA4P in cutting off tumor blood vessels. Together, the results demonstrated the feasibility of FT-TQT@FBS to assess the tumor vascular disruption, highlighting its potential application in evaluating the efficacy of vascular disrupting agents.

**Discussion**

Current medical imaging modalities used in clinical practice are mainly tomographic techniques, such as computed tomography (CT), magnetic resonance imaging (MRI), positron emission tomography (PET), and single-photon emission computed tomography (SPECT). However, the key constraints of aforementioned tomographic imaging modalities include hazardous ionizing radition, intrinsically limited spatiotemporal resolutions, and long imaging times[1]. In comparison, NIR-II fluorescence imaging has several advantages over other medical imaging modalities. It can provide fast feedback, nonionizing radiation and μm-scale resolution in wide-field imaging, although the penetration depth of NIR-II window is still significantly shallower than CT, PET and MRI[1]. Thus far, inorganic fluorophores, including carbon nanotubes, quantum dots, and lanthanide nanoparticles[15–22], have been widely used for preclinical animal NIR-II imaging. However, owing to the potential toxicity of heavy metals in inorganic fluorophores, it would be urgent to develop organic NIR-II dyes to facilitate FDA approval and clinical translation. However, most reported NIR-II organic fluorophores are hydrophobic and need to be encapsulated in a polymer matrix with slow in vivo clearance and impossible GMP (Good Manufacturing Practice) manufacture, which is far from clinical translation[42–45,56]. Due to tedious synthesis and purification, few studies have developed water-soluble NIR-II small-molecule dyes[3,10,23–25,40]. Nevertheless, water-soluble small-molecule dyes are still the leading candidates for clinical transformation, such as indocyanine green (ICG). Indocyanine green (ICG) was the first NIR fluorophore to be approved for use in humans. Thanks to the long emission tail in the NIR-II region (~1500 nm), ICG has been utilized for the first NIR-II imaging of liver-tumor patients recently, which promotes the clinical translation of NIR-II imaging[2,57]. However, clinically used ICG suffer from severe photobleaching and poor light stability[58]. Therefore, it is still a bottleneck for developing advanced NIR-II agents with excellent water-solubility, chem/photostability, longer absorption/emission wavelength, and good biocompatibility.

Molecular structural engineering of organic NIR-II fluorophores can provide powerful tunability for their optical properties[25]. In the last five years, D-A-D NIR-II dyes have been extensively explored for in vivo imaging[10,11,23–25,29,30,33]. To achieve better performance NIR-II D-A-D dyes, many efforts have been devoted to donor engineering[24,25,29,30]. However, it is noted that much less attention has been focused on the acceptor structures. In the present work, we reported a new electron acceptor, TQT, and confirmed that TQT had the highest alkali stability than BBT and PTQ. Meanwhile, in pH 8.5 PBS, TPA-TQT showed ultra-high stability than CH-4T, the reported NIR-II dyes based on BBT as acceptor. In addition, FT-TQT (0.49%) showed a higher quantum yield than FT-BBT (0.23%) and CH-4T (0.11%) in methanol. After loading FT-TQT into FBS, the relative

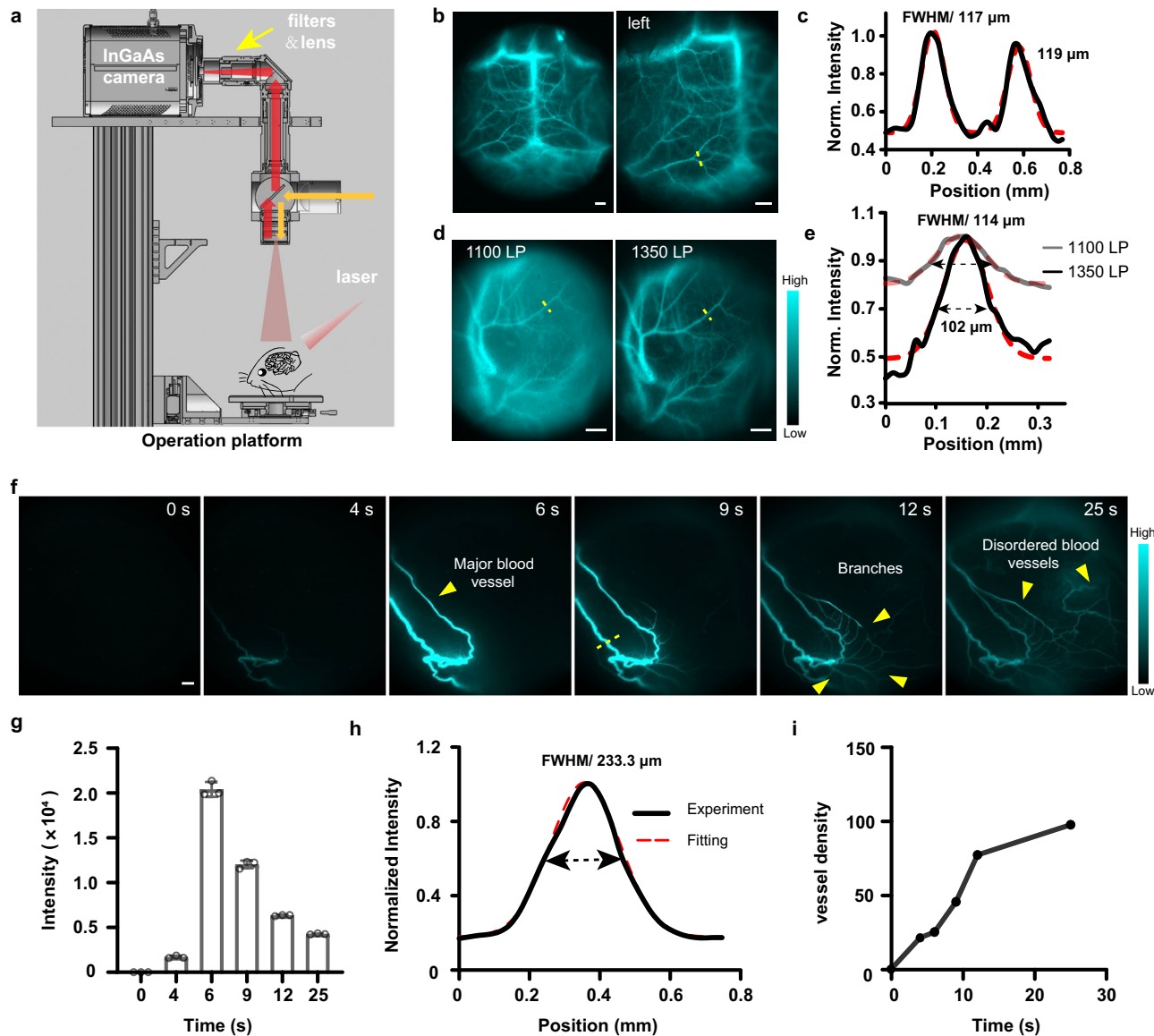

**Fig. 5 In vivo NIR-II imaging for the vascular network of brain and tumor. a** Scheme of zoom-stereo microscope NIR-II imaging system. **b** Representative NIR-II fluorescent images of cerebral vessels post intravenous injection of FT-TQT@FBS complexes. Scale bar: 1 mm. **c** Cross-sectional fluorescence intensity profile along the yellow line shown in panel (**b**). A Gaussian function fitted to the data (with FWHM) is also shown in red. **d** The tumor vessels of a mouse using FT-TQT@FBS complexes contrast at 1100 nm and 1350 nm LP filters. Scale bar: 1 mm. **e** Cross-sectional fluorescence intensity profiles (and Gaussian fits (red) with FWHM indicated by arrows) along the yellow lines in panel (**d**). **f** The recorded NIR-II video of tumor vasculatures in nude mice with xenograft osteosarcoma 143B at several time points p.i. of the FT-TQT@FBS complexes at 1100 nm LP filters. Scale bar: 1 mm. **g** The monitor of the fluorescence intensity profile of tumor vessels in NIR-II video imaging of **f**. Data are presented as mean ± s.d. derived from $n = 3$ independent measurements. **h** Cross-sectional intensity (solid black line) and Gaussian fit fluorescence intensity profiles (dotted red line) along the yellow lines in panel (**f**). **i** Quantitative analysis of the vascular density of tumor by using a vascular segmentation and quantification algorithm in panel (**f**). Norm, Normalized. The detailed imaging parameters for each image are listed in Supplementary Table 3.

fluorescence brightness of FT-TQT@FBS was 5-fold, 6.6-fold and 4-fold brighter than carbon nanotubes (QY = 0.04%; IR-26 = 0.05%), CH-PEG (QY = 0.03%; IR-26 = 0.05%) and IR-26 (QY = 0.05%), respectively[10,23], and was 8-fold enhancement in quantum yield than FT-TQT in water. Therefore, This work illustrates the feasibility of acceptor engineering strategy to develop better performance NIR-II molecular fluorophores with high stability and high brightness.

Real-time dynamic angiography that provides anatomical and hemodynamic information could deepen our understanding of vascular diseases and assess novel therapeutic approaches. Microscopic CT (micro-CT) and MRI are commonly used for imaging vascular structures. Nevertheless, the above-mentioned methods are limited by long scanning, limited spatial resolution, or high radiation dose and post-processing times[14]. For vascular hemodynamics, although laser Doppler can provide high temporal resolution, it can not accurately determine the diameter of blood vessels due to its poor spatial resolution and low contrast[14,59]. This study demonstrated that NIR-II imaging technique enables high-resolution structural vascular imaging, including the circulatory system, lymphatic drainage, and vascular network of the brain and tumor (Figs. 4, 5). A more prominent finding is that real-time monitoring of the tumor vascular disruption after treatment with combretastatin A4 phosphate

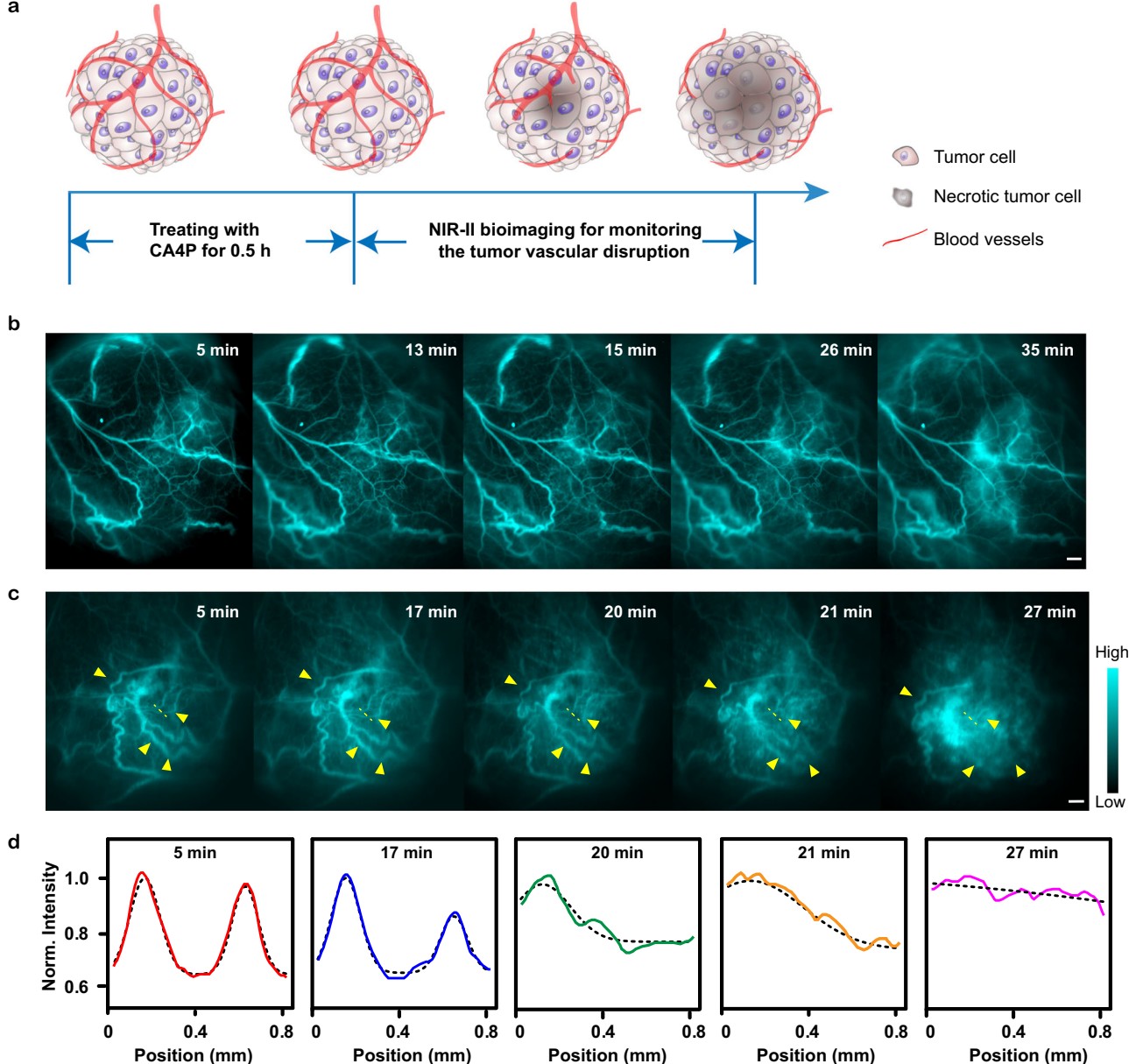

**Fig. 6 Real-time monitoring of the tumor vascular disruption after treatment with combretastatin A4 phosphate (CA4P). a** Schematic illustration of the tumor vascular disruption process after treatment with CA4P. **b** After 30 min administration of PBS, tumor vasculatures NIR-II imaging of nude mice with 143B xenograft osteosarcoma were obtained at different time points after FT-TQT@FBS p.i. **c** After 30 min administration of CA4P, tumor vasculatures NIR-II imaging of nude mice with xenograft osteosarcoma 143B was obtained at different time points after FT-TQT @FBS p.i. **d** Cross-sectional (solid line) and Gaussian fit fluorescence intensity profiles (dotted black line) of the vessel along the dashed yellow lines in **c** at different time points. Scale bar: 1 mm. Norm, Normalized. The detailed imaging parameters for each image are listed in Supplementary Table 3.

(CA4P) was identified by NIR-II fluorescent imaging for the first time (Fig. 6).

In the present work, the water-soluble probe FT-TQT shows superior imaging performance, but its clearance in vivo is relatively slow (Supplementary Fig. 17). The development of renal-clearable optical agents may be an effective approach to solve this problem, which can be quickly excreted from the body with little metabolic degradation. Renal clearance can be achieved by connecting organic fluorophores with water-soluble moieties, such as inulin, cyclodextrin, dextran, polyvinylpyrrolidone (PVP), or PEG[60]. Meanwhile, modifying organic fluorophores with targeted peptides or antibodies will further expand the application of these fluorophores.

In summary, this work describes a new electron acceptor—TQT, which shows better performance than BBT and PTQ. A small molecular organic NIR-II fluorophore, FT-TQT, achieves rapid, high resolution, and real-time vascular imaging in vivo. The optimized protein complexes, FT-TQT@FBS can be used for static or persistent imaging of dynamic vascular processes, including cerebral vasculature, tumor vessels, and tumor vascular disruption after treatment with CA4P. FT-TQT provides a powerful tool to monitor vascular-related diseases and evaluate the treatment efficacy of vascular-related therapy. The strategy on molecular engineering opens a new realm to design advanced NIR-II fluorophores based on TQT as an acceptor.

## Methods

**Synthesis**. For all synthesis details, please see the Supplementary Information.

**RP-HPLC measurement**. The basic stability of BBT-2Br, TQT-2Br, and PTQ-2Br were analyzed by RP-HPLC using eluent A and eluent B (eluent A: $H_2O$ containing 0.01% trifluoroacetic acid; eluent B: methanol. A/B = 10/90 for TQT-2Br, PTQ-2Br; A/B = 30/70 for BBT-2Br; 1 mL min$^{-1}$, detect wavelength: 254 nm; samples were filtered by 0.45 μm membrane). BBT-2Br, TQT-2Br and PTQ-2Br were dissolved in MeOH (containing 5% N,N-Dimethylformamide (DMF)). 1, 5, 10, 20 μL TEA, and 10 μL Trifluoroacetic acid (TFA) were added to the above solution, respectively. And make sure the mixture is at the same concentration. After incubation for 5 minutes, the mixture was subjected to RP-HPLC analysis, respectively.

**Cell culture**. In all, 143B (Human osteosarcoma cell) and 3T3 (mouse embryonic fibroblast cell) were cultured in DMEM containing high glucose (Gibco), all of which were supplemented with 10% FBS and 1% penicillin-streptomycin. The cells were expanded in tissue culture dishes and kept in a humidified atmosphere of 5% $CO_2$ at 37 °C. The medium was changed every other day. A confluent monolayer was detached with 0.5% trypsin and dissociated into a single-cell suspension for inoculation.

MTT assay ((3-(4,5-dimethylthiazol-2-yl)-2,5-diphenyl tetrazolium bromide) and all other reagents were purchased from Dalian Meilun Biotechnology Co., Ltd. Deionized water was obtained from a Millipore Milli-DI water purification system (Merck).

**Cytotoxicity of FT-TQT**. We determined the FT-TQT toxicity in vitro with MTT assay on 3T3/143B cells.

**Animal handling**. The following animal procedures were agreed with the Shanghai Experimental Animal Center of Chinese Academy of Sciences guidelines and performed under the institutional guidelines for animal handling. All operations related to animal experiments follow the relevant requirements of the Institutional Animal Care and Use Committee (IACUC) of Shanghai Institute of Material Medica, Chinese Academy of Sciences. In addition, 5 to 7-week-old BALB/c mice (15–20 g, ♀) were purchased from the Shanghai Experimental Animal Center of Chinese Academy of Sciences. Mice were housed under specific-pathogen-free conditions. The feeding environment is 25 °C, 35–45% humidity, 12 h light-dark alternation. All animals can drink water and eat freely. For xenograft osteosarcoma inoculation, 5 to 7-week-old nude mice were inoculated with five million 143B cells in 150 μL of the serum-free medium on the right leg. Tumors were allowed to grow for approximately 20 days before imaging. Before imaging, the mice were anesthetized using a rodent ventilator with air mixed with 4% isoflurane. The tail vein injection of contrast agents was carried out in the dark. The injected dose was a 200 μL bolus in a 1 × PBS (pH = 7.4) solution at a specified concentration. During the time course of imaging, the mouse was kept anesthetized by a nose cone delivering air mixed with 4% isoflurane. Mice were randomly selected from cages for all experiments. No blinding was performed.

**Whole-body vascular and lymphatic drainage NIR-II imaging**. Before whole-body vascular imaging, the hair of the abdominal region of Balb/c mice ($n = 3$) was removed by using depilatory gel. Mice were anesthetized and placed on a stage with a venous catheter for injection of FT-TQT (10 mg kg$^{-1}$). All whole-body NIR-II images were collected on MARS-FAST in vivo imaging system (Artemis Intelligent Imaging, Shanghai, China). Emission was typically collected with 1000, 1100, 1200, 1300, and 1400 nm LP filters (Thorlabs). For visualizing the collection of lymphatic drainage in the limb, FT-TQT (50 μL, 1 mg·mL$^{-1}$) was injected intradermally into the dorsal skin of the forepaws of the mice. Images were recorded at different time points post-injection. For hindlimb imaging, MARS-Pathfinder in vivo stereomicroscope imaging system (Artemis Intelligent Imaging, Shanghai, China) was performed to capture further high magnification.

**NIR-II imaging for the vascular network of the brain and tumor**. Brain vessel imaging was performed using MARS-Pathfinder in vivo stereomicroscope imaging system (Artemis Intelligent Imaging, Shanghai, China). Precisely, mice were placed on a stage with a venous catheter for i.v. injection of contrast and imaging agents. NIR-II imaging was performed with a variety of filters and exposure times. For brain vessel imaging, a 1400 LP filter was employed with a 10 s exposure time. For tumor vessels imaging, mice were imaged immediately after injecting FT-TQT@FBS (10 mg kg$^{-1}$). For assessing blood vessel development in the tumor, nude mice ($n = 3$) bearing subcutaneous 143B osteosarcoma were imaged using the NIR-II zoom-stereo microscope after injecting FT-TQT@FBS (10 mg kg$^{-1}$). Emission was typically collected with 1000, 1350 nm LP filter (Thorlabs).

**Real-time NIR-II imaging of tumor vascular disruption**. Nude mice ($n = 3$) bearing subcutaneous 143B osteosarcoma were anesthetized using a rodent ventilator with air mixed with 4% isoflurane. CA4P (200 μL, 1 mg mL$^{-1}$) was injected to induce a vascular disruption in the anesthetized state. After 30 min administration of CA4P, FT-TQT@FBS complexes were injected, and real-time NIR-II imaging was performed on tumor vascular.

**Vessel density characterization**. For Vessel Density Characterization details, please see the Supplementary Information.

**Density functional theory calculations for FT dyes**. Density function theory calculations were performed using Gaussian 03 revision C.02 software and B3LYP method and 6–31 G* basis set. The geometries were optimized using the default convergence criteria without any constraints.

**Statistical analysis**. Data were expressed as mean ± s.d. Differences between groups were assessed using unpaired two-tailed $t$ tests. *$P < 0.05$ was considered significant.

**Reporting summary**. Further information on research design is available in the Nature Research Reporting Summary linked to this article.

## Data availability
All data needed to evaluate the conclusions are present in the paper and the Supplementary information. The authors could provide additional data related to this paper with reasonable requestion. Source data is available for Figs. 2i, 3d, i, 4b, i, and 5g and Supplementary Figs. 7, 9, 10b, c, 11b, c, 13, 14, 15b, c, 16b, c, 17, and 22e in the associated source data file. Source data are provided with this paper.

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

## Acknowledgements

This work was partially supported by the National Science and Technology Innovation 2030 Major Project of China No. 2021ZD0203900 (H.C.), the Shanghai Municipal Science and Technology Major Project (Z.C., H.C.), the Lingang Laboratory, Grant No. LG-QS-202206-01 (H.C.), the National Natural Science Foundation of China under grant No. 82071976 (H.C.) and No. 81901799 (C.Q.), and the Shanghai Pujiang Program No. 19PJ1411100 (H.C.). We thank Prof. Cen Xie, Department of Drug Metabolism and Pharmacokinetics Research, the Affiliated Shanghai Institute of Materia Medica, for her generous help in cell culture.

## Author contributions

A.J., H.C., and Z.C. designed the study. H.L. and C.Q. jointly advised the study. A.J., W.L., and J.L. performed the synthesis. H.C., A.J., J.L., Y.W., and T.C measured and analysed the photophysics. H.C. built electronic instrumentation and optical configurations. H.L. and A.J. performed imaging experiments. A.J., H.L., Y.H., C.Q., and H.C. analysed images. H.L. performed cell culture experiments. A.J., Z.C., and H.C. wrote and edited the paper. All authors have approved the final version of the manuscript.

## Competing interests

The authors declare no competing interests.
