## [Peer Review File · Nature Communications]

Acceptor engineering for NIR-II dyes with high photochemical and biomedical performanceREVIEWER COMMENTS

Reviewer #1 (Remarks to the Author):

The authors developed a small molecule NIR-II dye with excellent water-solubility, chemical/photostability, and longer absorption/emission wavelength through discovering a novel electron acceptor, TQT. As compared with reported molecules including BBT and PTQ scaffolds used widely in NIR-II probes, TQT showed good stability in alkaline conditions for chemical modifications. The authors demonstrated the use of FT-TQT and FT-TQT@FBS in high-resolution structural vascular imaging, including circulatory system, lymphatic drainage and vascular network of brain and tumor. Real-time monitoring of tumor vascular disruption after treatment with combretastatin A4 phosphate (CA4P) was identified by NIR-II fluorescent imaging. As such, FT-TQT provides a powerful tool to monitor vascular-related diseases and evaluate treatment efficacy of vascular-related therapy. The TQT acceptor engineering strategy provides a promising approach to design next generation of NIR-II fluorophores for new biomedical applications. The experiments are well-designed for validating their claims and hypothesis such as water solubility, chemical stability, photostability, longer absorption/emission wavelength, higher SBR, deeper penetration (up to 7mm), good pharmacokinetic properties, and in vivo applications compatibility. Thus, I recommend its publication after addressing the following issues.

1. "A loading strategy with biocompatible fetal bovine serum (FBS) to the FT-TTQ was employed to improve the brightness further." How was the loading carried out? What ratio of probe to FBS was used? Was this complex properly characterized? The authors should provide proper details of preparation and characterization in the Methods section or SI.
2. Ex vivo biodistribution studies performed at 7 day and 14 days post-injection of FT-TQT indicated the probe was largely retained in, liver, spleen, lung, kidney and not cleared out as the FL intensity at 7 days and 14 days remained similar (Figure S11). A discussion can also be provided on how to improve the clearance of the probe without comprising on its use for real-time dynamic monitoring of vasculature changes.
3. Figure S11 showed ex vivo biodistribution studies to evaluate FT-TQT's distribution in vital organs including heart, liver, spleen, lung, kidney, tumor. However, the data for tumour is missing.
4. Ex vivo biodistribution studies to evaluate FT-TQT@FBS distribution in vital organs including heart, liver, spleen, lung, kidney, tumor should also be carried out as the FT-TQT@FBS complex should have different pharmacokinetic properties from the small molecule probe FT-TQT?
5. Statistical analysis should be carried out for Figure 4b to ensure differences are statistically significant.
6. The authors may consider citing other papers on NIR-II imaging and organ imaging (Nat. Rev. Mater., 2021, 6, 6, 1095–1113; Angew. Chem. Inter. Ed., 2020, 59, 11717; Angew. Chem. Inter. Ed., 2021, 60, 26454-).
7. There are several typos in the following sentences.
"For organic small molecule dyes, pursuing (pursuing) long absorption/emission wavelength, high chem/photostability, and high quantum yield are a subject of major interest²³." – Page 3
"It is labile and is decomposed in basic or reducing environment, restricting its' (its) use for development of more advanced NIR-II fluorophores^{21,38,39}." - Page 4
"Moreover, dyes with PTQ acceptor are all hydrophobic and need to be encapsulated in a polymer matrix which has slow in vivo clearance and does not favor for their potential clinical translation as well 40-43." – Page 4
"Generally, the biocompatibility of FT-TQT is excellent, showed (showing) a promising future for clinical translation." – Page 11
"Microscopic CT (micro-CT) and MRI are commonly used for inaging (imaging) vascular structures." – Page 17

Reviewer #2 (Remarks to the Author):

In this manuscript, Cheng et al. discussed a novel electron acceptor TQT, which shows higher stability than the widely used BBT and PTQ. The authors provide sufficient background with cited papers related to existing literature. As we know, two of the most challenging problems for most NIR-II dyes are solubility (in biological conditions) and low brightness. NIR-II fluorophores with high stability and brightness are sought-after for biomedical applications. The most significant novelty of the manuscript presumably was the good stability of TQT. The authors should provide more evidence to support their discovery and statement. The following are concerns with the manuscript:

1. First, as stated by the authors, there is some debate over the accurate quantum yield of IR-26 (0.05%-0.5%). However, 0.05% is the most popular accepted value. I strongly recommend using the quantum yield of IR26 (0.05%) as a reference instead of 0.5%. There is no need to use 0.5% to make the data look better.
2. For the D-A-D dyes, the authors claimed that the TQT was an excellent acceptor with high photochemical performance. However, the evidence provided in the text was insufficient. What's the stability of FT-TQT and TPA-TQT in the presence of biological nucleophiles or oxidants, such as amino acid, sulfhydryl, ROS/RNS...? What's the stability of FT-TQT and TPA-TQT in serum at 37°C?
3. Fig1, the authors claimed that TQT based D-A-D NIR-II dyes have "excellent stability in alkaline condition," detailed evidence was lacking.
4. Fig1, the authors claimed that TQT based D-A-D NIR-II dyes have "high brightness," no data (brightness = $\square * \square$) was provided in the text or SI.
5. The chemostability was tested with TPA-TQT, while the optical properties were tested and the imaging was performed with FT-TQT? Why? How about the chemostability of FT-TQT and the optical properties of TPA-TQT?
6. Fig 2d, what does the peak that appears around 8 min mean?
7. Fig2 j-I, did the data was collected immediately after getting the dye solution? How long is TPA-TQT stable in pH 8, pH 8.5 solutions or in a protic polar solvent such as MeOH? Hours or day?
8. FT-TQT was found to have a pretty long blood half-life time (~10 h). Why? Please provide explanations.
9. Line 366, "The optimized dye, FT-TQT@FBS can be used for static...", T-TQT@FBS is not a dye but a nanoparticle.
10. The physicochemical and optical properties of new fluorophores need to be analyzed in serum-containing warm media. Biodistribution (% ID/g) and quantification need to be considered with considering photo-decay (photo-ablation by laser), thickness and optical properties (scattering coefficient) of each organ, and other pharmacokinetic parameters.
11. Table S1, does the \square value was tested in MeOH or H₂O? Please provide \square and brightness values both in MeOH and H₂O.

Point-by-Point Response to Reviewers' Comments

We greatly appreciate the reviewers' valuable comments and allowing us to improve this manuscript. According to the comments, related revisions have been made in the revised manuscript. Below is the detailed information for our responses to reviewers' comments.

Reviewer #1 (Remarks to the Author):

The authors developed a small molecule NIR-II dye with excellent water-solubility, chemical/photostability, and longer absorption/emission wavelength through discovering a novel electron acceptor, TQT. As compared with reported molecules including BBT and PTQ scaffolds used widely in NIR-II probes, TQT showed good stability in alkaline conditions for chemical modifications. The authors demonstrated the use of FT-TQT and FT-TQT@FBS in high-resolution structural vascular imaging, including circulatory system, lymphatic drainage and vascular network of brain and tumor. Real-time monitoring of tumor vascular disruption after treatment with combretastatin A4 phosphate (CA4P) was identified by NIR-II fluorescent imaging. As such, FT-TQT provides a powerful tool to monitor vascular-related diseases and evaluate treatment efficacy of vascular-related therapy. The TQT acceptor engineering strategy provides a promising approach to design next generation of NIR-II fluorophores for new biomedical applications. The experiments are well-designed for validating their claims and hypothesis such as water solubility, chemical stability, photostability, longer absorption/emission wavelength, higher SBR, deeper penetration (up to 7mm), good pharmacokinetic properties, and in vivo applications compatibility. Thus, I recommend its publication after addressing the following issues.

Response: Thanks for the reviewer's valuable comments. We have revised our paper according to the comments.

Comments

1. *“A loading strategy with biocompatible fetal bovine serum (FBS) to the FT-TTQ was employed to improve the brightness further.” How was the loading carried out? What ratio of probe to FBS was used? Was this complex properly characterized? The authors should provide proper details of preparation and characterization in the Methods section or SI.*

Response: We appreciate the reviewer’s helpful suggestion. We have performed further studies and added the following contents in the **Supplementary Figure 6**. Briefly, 50 μg FT-TQT was dissolved in 100 μL 50%FBS and the solution was added in a sealed tube into 70-75⁰C water baths for 10 min. Finally, the solution was cooled to room temperature for *in vivo* NIR-II imaging. We also calculated the ratio of probe to FBS. Generally, the content of albumin is 30-50 g/L in FBS and bovine serum albumin (BSA) is the main form of albumin in FBS. So FBS is greatly excessive to the probe in this complex by calculation. Furthermore, we have tested the optical spectrum of this complex and its nanoparticle size. The results showed that FT-TQT, FT-TQT@FBS without Heated and FT-TQT@FBS Heated had the same maximum absorption peaks in deionized water. The emission of FT- TQT@FBS Heated ($\lambda_{\text{em}} = 999 \text{ nm}$) had 40 nm blue-shift than FT-TQT ($\lambda_{\text{em}} = 1039 \text{ nm}$) in the emission spectra. Meanwhile, the nanoparticle size of FT- TQT@FBS Heated was mainly distributed in 10-1000 nm.

Detailed preparation and characterization information was provided as follows,

Preparation of FT-TQT@FBS heated complexes.

*Briefly, 50 μg FT-TQT was dissolved in 100 μL 50%FBS and vortex the solution to mix evenly. Then the above solution was added in a sealed tube into 70-75⁰C water baths for 10 min. Finally, the solution was cooled to room temperature for *in vivo* NIR-II imaging.*

Figure S6. a) Normalized absorption spectrum of FT-TQT, FT-TQT@FBS without Heated and FT-TQT@FBS Heated in deionized water. b) Normalized fluorescence emission spectrum of FT-TQT, FT-TQT@FBS without Heated and FT-TQT@FBS Heated in deionized water. c) Photographs of the FBS, FBS-Heated and FT-TQT@FBS Heated in PBS solutions. d). The size of FBS, FBS-Heated and FT-TQT@FBS Heated by dynamic light scattering.

We have also added the following content in the revised manuscript:

Page 14, Line 6: *The preparation procedures and characterization of FT-TQT@FBS are described in the **Figure S6**.*

2. *Ex vivo biodistribution studies performed at 7 day and 14 days post-injection of FT-TQT indicated the probe was largely retained in, liver, spleen, lung, kidney and not cleared out as the FL intensity at 7 days and 14 days remained similar (Figure S11). A discussion can also be provided on how to improve the clearance of the probe without comprising on its use for real-time dynamic monitoring of*

vasculature changes.

Response: Thanks for the reviewer's valuable comments. We strongly agree with the reviewer's suggestions. We have provided a discussion on how to improve the clearance of the probe on **page 19**. Meanwhile, we have refined the paragraph in which the probe is used to monitor vasculature changes in real time.

We have added the following contents in the revised manuscript.

Page 19, Line 16: *“In the present work, the water-soluble probe FT-TQT shows superior imaging performance, but its clearance in vivo is relatively slow (Figure S13). The development of renal-clearable optical agents may be an effective approach to solve this problem, which can be quickly excreted from the body with little metabolic degradation. Renal clearance can be achieved by connecting organic fluorophores with water-soluble moieties, such as inulin, cyclodextrin, dextran, polyvinylpyrrolidone (PVP) or PEG⁶⁰. Meanwhile, modifying organic fluorophores with targeted peptides or antibodies will further expand the application of these fluorophores.”*

We have deleted **Page 19, Line 1-3:** *“The blood circulation system is involved in a variety of pathological processes, including peripheral arterial diseases (PADs), cerebrovascular diseases and tumor angiogenesis and metastasis.”* in our revised manuscript.

3. Figure S11 showed ex vivo biodistribution studies to evaluate FT-TQT's distribution in vital organs including heart, liver, spleen, lung, kidney, tumor. However, the data for tumour is missing.

Response: Thanks for the reviewer's valuable comments. We apologize for our mistake. The purpose of **Figure S13** (Previous **Figure S11**) is to prove that FT-TQT has good biocompatibility. Thus, the experiment was carried out on normal Balb/c mice without tumor (see **Page 13, Line 6** in manuscript, *“we evaluated in vivo toxicity by injecting FT-TQT into normal Balb/c mice”*). So, the data for the tumor was not provided. We have revised the incorrect expression in the revised Supporting Information and

manuscript. Considering photo-decay (photo-ablation by laser), thickness and optical properties (scattering coefficient) of each organ, and other pharmacokinetic parameters, we homogenized the tissues and requantified the *Ex vivo* biodistribution of FT-TQT in vital organs.

Figure S13. **a)** *Ex vivo* NIR-II imaging of the vital organs at 7 and 14 days post-injection. Histogram of the fluorescence intensities of the organs and tissues before **(b)** and after **(c)** homogenizing. Data as mean values \pm s.d. ($n = 3$). **d)** Representative major organ histology (H&E stained) of FT-TQT treated mice at 7 and 14 days post-injection. Scale bar: 100 μ m.

4. *Ex vivo* biodistribution studies to evaluate FT-TQT@FBS distribution in vital organs including heart, liver, spleen, lung, kidney, tumor should also be carried

out as the FT-TQT@FBS complex should have different pharmacokinetic properties from the small molecule probe FT-TQT?

Response: Thanks for the reviewer's valuable comments. We strongly agree with the reviewer's comments that FT-TQT and FT-TQT@FBS complex should have different pharmacokinetic properties. It is necessary to evaluate the biocompatibility of new NIR-II fluorophores. We first evaluated the *ex vivo* distribution of FT-TQT in vital organs of normal mice. In order to improve the brightness of FT-TQT, FT-TQT@FBS complex was prepared and used to monitor the changes of tumor blood vessels after drug treatment. Many reported NIR-II nanoprobes are used for tumor imaging. The aggregation of FT-TQT@FBS in tumor is not the research focus of this manuscript. Thus, to compare the *ex vivo* distribution differences of FT-TQT and FT-TQT@FBS in vital organs, including heart, liver, spleen, lung, and kidney, we did the same experiment by injecting FT-TQT@FBS (heated) into normal Balb/c mice, and the results are as follows. Compared with FT-TQT, FT-TQT@FBS is most widely distributed in the liver, which has a large particle size and is hard to excrete through the kidney. (please see *Questions #3* for details). However, FT-TQT is mainly distributed in the kidney, followed by the liver.

Continued on Next Page

Figure S14. Toxicity and biodistribution assay of FT-TQT@FBS (100 μg , 200 μL 50%FBS). **a)** *Ex vivo* NIR-II imaging of the vital organs at 7 and 14 days post-injection. Histogram of the fluorescence intensities of the organs and tissues before (**b**) and after (**c**) homogenizing. Data as mean values \pm s.d. ($n = 3$). **d)** Representative major organ histology (H&E stained) of FT-TQT@FBS treated mice at 7 and 14 days post-injection. Scale bar: 100 μm . (Laser, 808 nm; Power, 90 $\text{mW}\cdot\text{cm}^{-2}$; Exposure time, 30 ms; 1000 nm Long Pass filter)

Meanwhile, we have added the following contents in the revised manuscript,

Page14, Line 15: *To evaluate the biocompatibility of FT-TQT@FBS, NIR-II imaging of the vital organs was performed after intravenous administration at 7 and 14 days*

(**Figure S14**). Compared with FT-TQT, FT-TQT@FBS is most widely distributed in the liver, which has a large particle size and is hard to excrete through the kidney. However, FT-TQT is mainly distributed in the kidney, followed by the liver. Additionally, the H&E staining of the major organs indicated no noticeable pathological change after FT-TQT@FBS treatment.

5. *Statistical analysis should be carried out for Figure 4b to ensure differences are statistically significant.*

Response: Thanks for the reviewer's valuable comments. We have performed the statistical analysis in Figure 4b by unpaired two-tailed *t* tests and *P* value was less than 0.0001, which indicated that there was a significant difference between the two groups. We have revised the manuscript on **page 38** accordingly.

6. *The authors may consider citing other papers on NIR-II imaging and organ imaging (Nat. Rev. Mater., 2021, 6, 6, 1095 - 1113; Angew. Chem. Inter. Ed., 2020, 59, 11717; Angew. Chem. Inter. Ed., 2021, 60, 26454-).*

Response: Thanks for the reviewer's valuable comments. We have cited the above reference as Reference 5, 13, and 60 in the revised manuscript.

7. *There are several typos in the following sentences.*

“ For organic small molecule dyes, pursueing (pursuing) long absorption/emission wavelength, high chem/photostability, and high quantum yield are a subject of major interest²³. ” - Page 3

“It is labile and is decomposed in basic or reducing environment, restricting its’ (its) use for development of more advanced NIR-II fluorophores^{21,38,39}. ” - Page 4

“Moreover, dyes with PTQ acceptor are all hydrophobic and need to be encapsulated in a polymer matrix which has slow in vivo clearance and does not favor for their potential clinical translation as well 40-43. ” - Page 4

“Generally, the biocompatibility of FT-TQT is excellent, showed (showing) a promising future for clinical translation. ” - Page 11

“Microscopic CT (micro-CT) and MRI are commonly used for inaging (imaging) vascular structures. ” - Page 17

Response: Thanks so much for the reviewer's comments. We have polished our manuscript carefully and corrected the grammatical, styling, and typos found in our manuscript. Detailed modifications are also provided as follows:

Page 3, line 17: "pursueing" was changed to "pursuing".

Page 4, line 20: "restricting its' use" was changed to "restricting its use".

Page 5, line 4: "Moreover, dyes with PTQ acceptor are all hydrophobic and need to be encapsulated in a polymer matrix which *has* slow in vivo clearance and *does* not favor for their potential clinical translation as well" was changed to "Moreover, dyes with PTQ acceptor are all hydrophobic and need to be encapsulated in a polymer matrix which *have* slow in vivo clearance and *do* not favor for their potential clinical translation as well"

Page 13, line 15: "showed" was changed to "showing".

Page 19, line 6: "inaging" was changed to "imaging".

Additionally, we have highlighted other modifications in the revised manuscript.

Reviewer #2 (Remarks to the Author)

In this manuscript, Cheng et al. discussed a novel electron acceptor TQT, which shows higher stability than the widely used BBT and PTQ. The authors provide sufficient background with cited papers related to existing literature. As we know, two of the most challenging problems for most NIR-II dyes are solubility (in biological conditions) and low brightness. NIR-II fluorophores with high stability and brightness are sought-after for biomedical applications. The most significant novelty of the manuscript presumably was the good stability of TQT. The authors should provide more evidence to support their discovery and statement. The following are concerns with the manuscript:

Response: Thanks for the reviewer's valuable comments. We have revised the manuscript according to the comments.

Comments

1. First, as stated by the authors, there is some debate over the accurate quantum yield of IR-26 (0.05%-0.5%). However, 0.05% is the most popular accepted value. I strongly recommend using the quantum yield of IR26 (0.05%) as a reference instead of 0.5%. There is no need to use 0.5% to make the data look better.

Response: Thanks for the reviewer's valuable comments. We strongly agree with it. Simultaneously, we consulted the relevant reference and taken $IR_{26} = 0.05\%$ as the standard (*The journal of physical chemistry letters*, 2010, 1(16): 2445-2450.).

We have made corresponding modifications as follows:

Page 6, Line 5: *FT-TQT (0.49%) showed a higher quantum yield than FT-BBT (0.23%) in methanol.*

Page 6, Line 10: *... with FT-TQT in PBS ($QY_{FT-TQT@FBS} = 0.2\%$, $QY_{FT-TQT} = 0.025\%$).*

Page 10, Line 22: *NIR-II dye CH-4T were determined to be 0.49%, 0.23%, and 0.11% in methanol.*

Page 11, Line 2: *with IR-26 as a reference, $QY = 0.05\%$*

Page 18, Line 15: *FT-TQT (0.49%) showed a higher quantum yield than FT-BBT (0.23%) and CH-4T (0.11%) in methanol.*

Table S1. Quantum yield and brightness values of FT-BBT, FT-TQT, and FT-TQT@FBS complexes in MeOH and H₂O.

Fluorophore	QY_{MeOH}	QY_{H_2O}	ϵ_{H_2O}	ϵ_{MeOH}	$\epsilon^*\Phi_{H_2O}$	$\epsilon^*\Phi_{MeOH}$
	(%)	(%)	($\times 10^3 M^{-1} \cdot cm^{-1}$)	($\times 10^3 \cdot M^{-1} \cdot cm^{-1}$)	($M^{-1} \cdot cm^{-1}$)	($M^{-1} \cdot cm^{-1}$)
FT-BBT	0.23	---	11.4	16.25	---	37.4.
FT-TQT	0.49	0.025	8.94	10.66	2.24	52.2
CH-4T	0.11	0.06	3.03	9.88	1.82	10.87
FTTQT@FBS	---	0.20	---	---	---	---

2. *For the D-A-D dyes, the authors claimed that the TQT was an excellent acceptor with high photochemical performance. However, the evidence provided in the text was insufficient. What's the stability of FT-TQT and TPA-TQT in the presence of biological nucleophiles or oxidants, such as amino acid, sulfhydryl, ROS/RNS...? What's the stability of FT-TQT and TPA-TQT in serum at 37⁰C?*

Response: Thanks for the reviewer's valuable suggestions. We also believe that stability is an important property of dyes. Therefore, to examine the chemical stability of BBT and TQT based dyes in the presence of reactive oxygen/nitrogen species (ROS/RNS), metal ions, and active biomolecules, we measured the absorption spectra of CH-4T, FT-BBT, TPA-TQT and FT-TQT after incubated with the above substances at 37⁰C for 1 hour (**Figure S17-19**). The results showed that all dyes are stable in the presence of metal ions (K⁺, Na⁺, Ca²⁺, Mg²⁺, Fe²⁺, and Zn²⁺) and active biomolecules (GSH, Cys, Hcy, ascorbic acid (AA), and dehydroascorbic acid (DHA)), although CH-4T is red-shifted in the presence of Ga²⁺ and Fe²⁺. However, BBT-based dyes have shown poor stability than TQT-based dyes in the presence of ROS/RNS, especially ClO⁻. Meanwhile, TPA-TQT and FT-TQT remained stable after incubation in water and mouse serum at 37⁰C for 24 hours. At the same time, it also has good photostability (**Figure S20**). These results show that TQT-based D-A-D dyes hold high photochemical performance in the presence of biological nucleophiles or oxidants and remain stable in mouse serum at 37⁰C.

We have added the following contents in the revised supporting information.

As shown in Figure S17, CH-4T, TPA-TQT, FT-BBT, and FT-TQT remain stable under the condition of coincubation of biological nucleophilic molecules, such as AA, DHA, Cys, Hcy, and GSH.

Figure S17. Changes of absorption curves of dyes (10 μM), CH-4T (a), TPA-TQT (b), FT-BBT (c) and FT-TQT (d) after reaction with active biomolecules. ascorbic acid, AA; dehydroascorbic acid, DHA; L-Cysteine, Cys; Glutathione, GSH; DL-Homocysteine, Hcy. Incubation conditions: 37⁰C water bath for one hour.

As shown in **Figure S18**, CH-4T, which is based on BBT acceptor, degrades in varying degrees under the coincubation of a variety of RNS/ ROS active molecules, except for hydrogen oxide. Additionally, nearly half of the decomposition of FT-BBT also occurred under the condition of co-incubation with NaClO. On the contrary, TPA-TQT and FT-TQT still remain stable under the condition of coincubation of these active molecules.

Figure S18. Changes of absorption curves of CH-4T (a), TPA-TQT (b), FT-BBT (c), and FT-TQT (d) after reaction with ROS/RNS. Incubation conditions: 37⁰C water bath for one hour.

As shown in **Figure S19**, in the presence of Ca^{2+} and Fe^{2+} , the absorption spectrum of CH-4T is red-shifted, while the other three dyes are still stable after coincubation with various metal ions.

Continued on Next Page

Figure S19. Changes of absorption curves of dyes (10 μM), CH-4T (a), TPA-TQT (b), FT-BBT (c) and FT-TQT (d) after reaction with metal ions (1 mM). Incubation conditions: 37 $^{\circ}\text{C}$ water bath for one hour.

As shown in **Figure S20**, TPA-TQT and FT-TQT remained stable after incubation in water and mouse serum at 37 $^{\circ}\text{C}$ for 24 hours. At the same time, it also has good photostability. The fluorescence intensity of FT-TQT was 9.6 times higher than that of TPA-TQT in H₂O at the same concentration. The fluorescence intensity of TPA-TQT increased by 14.5 times, and that of FT-TQT increased by 8.2 times after incubated with serum for 1 hour at 37 $^{\circ}\text{C}$.

Continued on Next Page

Figure S20. The stability of TPA-TQT and FT-TQT in mouse serum. The absorption spectra of TPA-TQT in H₂O (a) and mouse serum (b) at different times; The absorption spectra of FT-TQT in H₂O (c) and mouse serum (d) at different times; e) The fluorescence intensity of TPA-TQT and FT-TQT with the same concentration (8.5 μ M) in water and mouse serum at 37°C for 1 hour (1000 LP, 100 ms, 100 mW· cm⁻²) f) Photostability of TPA-TQT and FT-TQT in serum under continuous laser irradiation (808 nm, 100 mW· cm⁻²).

Meanwhile, we have added the following contents in the revised manuscript.

Page 8, Line 22: *To examine the chemical stability of BBT and TQT based dyes in the presence of reactive oxygen/nitrogen species (ROS/RNS), metal ions, and active*

biomolecules, we measured the absorption spectra of CH-4T, FT-BBT, TPA-TQT and FT-TQT after incubated with the above substances at 37°C for 1 hour (**Figure S17-19**). The results showed that all dyes are stable in the presence of metal ions (K^+ , Na^+ , Ca^{2+} , Mg^{2+} , Fe^{2+} , and Zn^{2+}) and active biomolecules (GSH, Cys, Hcy, ascorbic acid (AA), and dehydroascorbic acid (DHA)), although CH-4T is red-shifted in the presence of Ga^{2+} and Fe^{2+} . However, BBT-based dyes have shown poor stability than TQT-based dyes in the presence of ROS/RNS, especially ClO^- .

Page 10, Line 4: Besides, TPA-TQT and FT-TQT showed ultra-high photochemical stability in methanol and mouse serum (**Figure S20, S25**). However, the fluorescence intensity of FT-TQT was 9.6 times higher than that of TPA-TQT in H_2O at the same concentration (**Figure S20e**). Meanwhile, the fluorescence intensity of FT-TQT increased 8.2 times after incubated with mouse serum at 37°C for 1 hour.

3. Fig1, the authors claimed that TQT based D-A-D NIR-II dyes have “excellent stability in alkaline condition,” detailed evidence was lacking.

Response: Thanks for the reviewer's valuable comments. We have added the following experiments to provide more evidence.

In **Figure 2d, 2e**, and **Figure S16** (previous **Figure S13**), the stability of BBT, TQT, and PTQ in alkali solution with different triethylamine volume ratios were tested by HPLC. The results showed that TQT is the most stable acceptor. To investigate the photochemical properties of TQT based on D-A-D NIR-II dyes further, TPA-TQT was synthesized, whose structure is similar to CH-4T, a BBT based dye (**Figure 2f**). In **Figure 2i – 2l**, we demonstrated the stability of TPA-TQT and CH-4T under physiological alkaline conditions (pH 7.4, 8.0, and 8.5) after 808 nm laser irradiation. TPA-TQT in different pH solutions remained stable under continuous 808 nm laser irradiation for 30 minutes; While the color of CH-4T solution under pH 8.5 changed from green to yellow after three minutes of illumination, and the changes in the absorption curve indicating that TPA-TQT is more alkaline resistant than CH-4T.

In the revised manuscript, we evaluated the alkali stability of BBT and TQT-based dyes in different alkali solutions (**Figure S21**) and in NaOH solutions with different mass concentrations (**Figure S22**). The results showed that TQT-based NIR-II dyes, TPA-TQT and FT-TQT showed the best stability in all tested alkali solutions. Meanwhile, BBT and TQT-based dyes were incubated at pH 5.0 - 10.0 with 37°C water baths (**Figure S23**). Nearly half of CH-4T decomposed when incubated under pH 8.5 for 24 hours. TQT-based dyes remained stable after incubated for 48 h at pH 5.0-10.0.

We have added the following contents in the Supplementary Materials to provide more detailed evidence.

*As shown in **Figure S21**, BBT based NIR-II dye, CH-4T showed the worst stability in different alkali solutions, including 1%NaOH, 1%TEA, and 1%DIEA. Another BBT-based dye, FT-BBT, showed higher stability in alkali solutions than CH-4T, which illustrates that the electron donor unit plays a role in NIR-II dyes' stability. However, FT-BBT is still partially decomposed in 1%NaOH. Undoubtedly, TQT-based NIR-II dyes, TPA-TQT and FT-TQT showed the best stability in all tested alkali solutions.*

Continued on Next Page

Figure S21. The stability of CH-4T (a), FT-BBT (b), TPA-TQT (c), and FT-TQT (d) in different alkaline aqueous solutions at room temperature. Dye concentration: 50 μM ; 1% represents mass concentration (g / 100 mL or mL / 100 mL). Sodium hydroxide, NaOH; Triethylamine, TEA; N, N-Diisopropylethylamine, DIEA; Potassium carbonate, K_2CO_3 .

Additionally, we evaluated the changes of absorption curves of FT-BBT, TPA-TQT, and FT-TQT in NaOH solutions with different mass concentrations.

As shown in **Figure S22**, in 5% NaOH solution, FT-BBT degraded 42% after incubated for 24 hours. Excitedly, TQT-based dyes remain stable even if in 5%NaOH.

Figure S22. The stability of FT-BBT, TPA-TQT, and FT-TQT in different mass concentrations of NaOH at room temperature for different times. Dye concentration: 50 μM .

These results show that TQT-based D-A-D dyes can be widely used for chemical modification under different alkali conditions. We further evaluated the stability of these fluorophores at pH 5.0-10.0 with 37°C water baths.

As shown in **Figure S23**, nearly half of CH-4T decomposed when incubated under pH 8.5 for 24 hours. FT-BBT degraded slightly under alkaline conditions. However, TQT-based dyes remained stable after incubated for 48 h at pH 5.0-10.0.

Figure S23. The stability of CH-4T (a), FT-BBT (b), TPA-TQT (c), and FT-TQT (d) under different pH conditions. Incubation conditions: 37⁰C water bath. [CH-4T, FT-BBT, FT-TQT]: 2 μ M; [TPA-TQT]: 6 μ M.

We have added the following contents in the revised manuscript,

Page 9, Line 9: To evaluate the alkali stability of the four fluorophores, firstly, the absorption spectra of four fluorophores were tested at different time points in different alkali solutions (Figure S21). CH-4T showed the worst stability in different alkali solutions, including 1%NaOH, 1%TEA, and 1%DIEA. It was degraded by 98%, 76% and 70% in 1%NaOH, 1%TEA and 1%DIEA on day seven, respectively. Another BBT-based dye, FT-BBT showed higher stability in alkali solutions than CH-4T, which

illustrates the electron donor unit plays a role in the stability of NIR-II dyes. However, FT-BBT still partially decomposed in 1%NaOH (28% on the seventh day). Undoubtedly, TQT-based NIR-II dyes, TPA-TQT and FT-TQT showed the best stability in all tested alkali solutions. Additionally, we evaluated the changes of absorption curves of FT-BBT, TPA-TQT, and FT-TQT in NaOH solutions with different mass concentrations (**Figure S22**). In 5% NaOH solution, FT-BBT degraded 42% after incubated for 24 hours. Excitedly, TQT-based dyes remain stable even if in 5%NaOH, which shows TQT-based D-A-D dyes can be widely used for chemical modification under different alkali conditions. CH-4T, FT-BBT, TPA-TQT, and FT-TQT were incubated at pH 5.0 - 10.0 with 37°C water baths (**Figure S23**). Nearly half of CH-4T decomposed when incubated under pH 8.5 for 24 hours.

We have revised “In comparison to its counterpart CH-4T using BBT as an acceptor, which is a widely used NIR-II dye, TPA-TQT showed ultra-high stability in pH 8.5 solutions” into “In comparison to its counterpart CH-4T using BBT as an acceptor, which is a widely used NIR-II dye, TPA-TQT showed ultra-high stability in the presence of reactive oxygen/nitrogen species (ROS/RNS), metal ions and active biomolecules and various alkali conditions” on **Page 5** in the revised manuscript.

4. Fig1, the authors claimed that TQT based D-A-D NIR-II dyes have “high brightness,” no data (brightness = $\epsilon \cdot \phi$) was provided in the text or SI.

Response: Thanks for the reviewer's valuable comments. We have added the following contents in the revised manuscript and SI.

As shown in **Table S1**, compared with FT-BBT ($37.4 \text{ M}^{-1} \cdot \text{cm}^{-1}$) and CH-4T ($10.87 \text{ M}^{-1} \cdot \text{cm}^{-1}$), FT-TQT ($52.2 \text{ M}^{-1} \cdot \text{cm}^{-1}$) hold high brightness in methanol. Meanwhile, FT-TQT ($2.24 \text{ M}^{-1} \cdot \text{cm}^{-1}$) has a slightly higher brightness than CH-4T in H_2O ($1.82 \text{ M}^{-1} \cdot \text{cm}^{-1}$). So, in **Fig 1**, we have changed “high brightness” into “high brightness than the counterpart with BBT as acceptor”.

Table S1. Quantum yield and brightness values of FT-BBT, FT-TQT, and FT-TQT@FBS complexes in MeOH and H₂O.

Fluorophore	QY _{MeOH}	QY _{H₂O}	ε _{H₂O}	ε _{MeOH}	ε*Φ _{H₂O}	ε*Φ _{MeOH}
	(%)	(%)	(x10 ³ M ⁻¹ ·cm ⁻¹)	(x10 ³ ·M ⁻¹ ·cm ⁻¹)	(M ⁻¹ ·cm ⁻¹)	(M ⁻¹ ·cm ⁻¹)
FT-BBT	0.23	---	11.4	16.25	---	37.4.
FT-TQT	0.49	0.025	8.94	10.66	2.24	52.2
CH-4T	0.11	0.06	3.03	9.88	1.82	10.87
FT-TQT@FBS	---	0.20	---		---	---
IR-26 (DCE)	0.05					

5. The chem-stability was tested with TPA-TQT, while the optical properties were tested and the imaging was performed with FT-TQT? Why? How about the chem-stability of FT-TQT and the optical properties of TPA-TQT?

Response: Thanks for the reviewer's valuable comments. To validate the stability of TQT based D-A-D dyes, we chose a widely used D-A-D dye, CH-4T, as a contrast, which is based on a classical electron acceptor BBT. Meanwhile, we synthesized its counterpart TPA-TQT, and preliminarily compared their UV-Vis, fluorescence spectra, and photostability under continuous 808 nm laser irradiation (**Figure 2f-2h**). We found that TPA-TQT showed the maximum absorption peak ($\lambda_{\text{max}} = 678$ nm) and the emission wavelength (946 nm) have blue-shifted compared to CH-4T. Generally, this is unfavorable for NIR-II imaging. Long emission wavelengths usually have lower tissue auto-fluorescence, photo-absorption and photo-scattering, and higher penetration. Therefore, we focused on the chemical stability of TPA-TQT and CH-4T. Of course, another study focusing on the optical properties and biological applications of TPA-TQT is also in progress.

In the previous manuscript, we demonstrated the stability of pH 7.4, 8.0, and 8.5 under physiological alkaline conditions after 808 nm laser irradiation (**Figure 2i-2l**). TPA-TQT in different pH solutions remained stable under continuous 808 nm laser irradiation for 30 minutes; While the color of CH-4T solution under pH 8.5 changed from green to yellow after three minutes of illumination, and the changes in the absorption curve indicating that TPA-TQT is more alkaline resistant than CH-4T. However, our previous data is still insufficient.

Therefore, in the revised manuscript and supplementary materials, we first tested the stability of CH-4T and TPA-TQT under 37⁰C water bath with different pH (5.0-10.0) over time (**Figure S23, S24**). The results showed that TPA-TQT remained stable until 96 h, while nearly half of CH-4T decomposed after incubated for 24 hours at pH 8.5. Then, we tested the stability of bioactive molecules and oxidants, such as ROS/RNS, metal ions, and mercaptoamino-acids. The results also showed that TPA-TQT had better stability than CH-4T (**Figure S17-19**). Finally, we measured their stability (37⁰C) in different alkali solutions (**Figure S21**). CH-4T showed the worst stability in different alkali solutions, including 1%NaOH, 1%TEA, and 1%DIEA. It was degraded by 98%, 76%, and 70% in 1%NaOH, 1%TEA, and 1%DIEA on day seven, respectively. In comparison, TPA-TQT showed the best stability in all tested alkali solutions. The above results confirm that TPA-TQT has better chemical stability than CH-4T, which means that the TQT-based D-A-D dye shows better stability than the BBT-based D-A-D dyes.

To develop red-shifted TQT fluorescent dyes, FT-TQT and its counterpart FT-BBT were synthesized. In the manuscript, we reported in detail that FT-TQT has better optical imaging performance than FT-BBT (**Figure 4, 5**). In the modified supplementary materials, we tested the chemical stability of FT-TQT and FT-BBT. These results proved that FT-TQT also has high chemical stability and alkali stability. At the same time, it has good stability in aqueous solution, polar solvent (methanol), and mouse serum (**Figure S17-25**). (please see *Questions #2, #3 and #6* for details)

The following content has been added in the revised manuscript:

Page 7: Compared with CH-4T, TPA-TQT has a significant blue shift, which is unfavorable for NIR-II imaging.

Page 10, Line 9: Thus, considering the emission wavelength and stability, we chose FT-TQT and FT-BBT as the next research object to compare the differences of optical properties between BBT- and TQT-based dyes.

6. Fig 2d, what does the peak that appears around 8 min mean?

Response: Thanks for the reviewer's valuable comments. This is an impurity substance in the sample. We have calculated the relative peak area of two peaks under different alkali conditions. The content of the impurity peak is less than 5% except sample **4**, which may be caused by the fluctuation of the instrument.

Name	Relative peak area	
	Peak 1	Peak 2
1 (control)	$t = 8.64$ min; 4.77%	$t = 12.097$ min; 95.23%
2 (excess TFA)	$t = 8.737$ min; 3.94%	$t = 12.273$ min; 96.06%
3 (1 μ L TEA)	$t = 8.647$ min; 4.98%	$t = 12.087$ min; 95.02%
4 (5 μ L TEA)	$t = 8.567$ min; 5.12%	$t = 11.987$ min; 94.88%
5 (10 μ L TEA)	$t = 8.61$ min; 2.70%	$t = 12.033$ min; 97.30%
6 (20 μ L TEA)	$t = 8.62$ min; 4.19%	$t = 12.067$ min; 95.81%

To avoid confusing readers and provide high-quality images, we re-purified the TQT-

2Br and monitored its stability under different TEA volume ratios by HPLC. The results are as follows.

Fig. 2d HPLC chromatograms of TQT-2Br at various acid-base conditions. Bright field images of TQT-2Br in MeOH (5% DMF, 1 mL) at various acid-base conditions. 1: control solution; 2: control solution + excess trifluoroacetic acid (TFA); 3: control solution + 1 μ L Triethylamine (TEA); 4: control solution + 5 μ L Triethylamine (TEA); 5: control solution + 10 μ L Triethylamine (TEA); 6: control solution + 20 μ L Triethylamine (TEA); Blank': 50 μ L DMF+20 μ L Triethylamine +930 μ L MeOH.

7. Fig2 j-I, did the data was collected immediately after getting the dye solution? How long is TPA-TQT stable in pH 8, pH 8.5 solutions or in a protic polar solvent such as MeOH? Hours or day?

Response: Thanks for the reviewer's valuable comments. In **Fig 2j-I**, the data was collected after continuous 808 nm laser irradiation for 30 minutes. The results showed that TPA-TQT in different pH solutions remained stable, while the color of CH-4T solution under pH 8.5 changed from green to yellow after three minutes of illumination. In the revised manuscript, TPA-TQT and FT-TQT remained stable after incubated at 37⁰C water baths in pH 8.0 and pH 8.5 for 96 h (**Figure S24**). Besides, TPA-TQT and FT-TQT showed ultra-high photochemical stability in methanol for 7 days (**Figure S25**).

We have added the following contents in the **SI**,

As shown in **Figure S24**, TPA-TQT and FT-TQT remain stable in pH 8.0, 8.5 for 96 h at 37°C. Meanwhile, they were also stable in MeOH at room temperature for seven days (**Figure S25**).

Figure S24. The stability of TPA-TQT (a) and FT-TQT (b) in pH 8.0, 8.5 for 96 h at 37°C. [TPA-TQT]:6 μ M; [FT-TQT]:2 μ M.

Figure S25. The stability of TPA-TQT and FT-TQT in MeOH at different time points. [TPA-TQT]:10 μ M; [FT-TQT]:5 μ M.

Additionally, we have added the following contents in the revised manuscript,

Page 10, Line 3: However, TPA-TQT and FT-TQT remained stable after incubated at pH 8.0 and pH 8.5 for 96 h (**Figure S24**). Besides, TPA-TQT and FT-TQT showed ultra-high photochemical stability in methanol and mouse serum (**Figure S20, S25**).

8. FT-TQT was found to have a pretty long blood half-life time (~10 h). Why? Please

provide explanations.

Response: Thanks for the reviewer's valuable comments. According to the literature, a common method to prolong the blood half-life of drugs is to attach drugs to blood components that have a long half-life by covalent binding or fusion, such as albumin. Because albumin can be recycled back into the blood by the neonatal Fc receptor, thus prolonging the blood half-life of the drug (*Bioconjugate chemistry*, 2016, 27(10): 2239-2247.). Evans Blue (EB), an albumin-binding dye, is often attached to drugs to prolong the plasma half-life of drugs (*Theranostics*, 2016, 6(2): 243). The FDA-approved magnetic resonance imaging (MRI) contrast agent gadofosveset trisodium (abrarvar) can reversibly bind serum albumin, resulting in more prolonged blood circulation than other gadolinium-based contrast agents (*Magn. Reson. Med.* 2015, 73, 244–53; *Vasc. Health Risk Manag.* 2008, 4, 1–9.)

D-A-D NIR-II dyes can be fused with proteins, such as FBS, HSA, and BSA (*Nature communications*, 2017, 8(1): 1-11; *Advanced Functional Materials*, 2020, 30(6): 1906343; *Sci. Adv.* 2019; 5 : eaaw0672.), and show the high imaging performance.

In the present work, as shown in **Figure S4** and **Figure S20e**, FT-TQT can bind to various proteins to enhance fluorescence intensity. FT-TQT showed higher fluorescence enhancement in mouse serum than in blood cells at room temperature. Meanwhile, the fluorescence of FT-TQT increased 8.2 times after incubated with mouse serum at 37°C for 1 h. These results indicated that FT-TQT could bind to protein components in mouse serum. Thus, we think that the interaction between FT-TQT and serum proteins is the main reason for its prolonged plasma half-life.

We have added the following contents on **Page 12, Line 14** in the revised manuscript:

Additionally, FT-TQT was found to have a pretty long blood half-life time (~10 h), which may cause by the interaction between FT-TQT and serum proteins, such as albumin⁴⁶. (Ref. 46: Jacobson, O., Kiesewetter, D. O., & Chen X. Albumin-binding Evans blue derivatives for diagnostic imaging and production of long-acting

therapeutics. Bioconjugate chem., 27: 2239-2247 (2016)

9. Line 366, “The optimized dye, FT-TQT@FBS can be used for static...”, FT-TQT@FBS is not a dye but a nanoparticle.

Response: Thanks for the reviewer's valuable comments. We have made a corresponding modification in the revised manuscript.

Page 20, Line 5; “dye” was changed to “protein complexes”.

10. The physicochemical and optical properties of new fluorophores need to be analyzed in serum-containing warm media. Biodistribution (% ID/g) and quantification need to be considered with considering photo-decay (photo-ablation by laser), thickness and optical properties (scattering coefficient) of each organ, and other pharmacokinetic parameters.

Response: Thanks for the reviewer's valuable comments. The chem-stability of TPA-TQT and FT-TQT in mouse serum and other media at 37°C was measured for 24 h, and the results were shown as **Figure S17** in the supporting information. The absorption spectra of TPA-TQT and FT-TQT remained unchanged after incubated with mouse serum and deionized water at 37°C for 24 h, illustrating the stability of TPA-TQT and FT-TQT in serum and deionized water. Meanwhile, the fluorescence intensity of TPA-TQT and FT-TQT remained stable after continuous 808 nm laser irradiation for 1 h in mouse serum and PBS (**Figure S2, Figure S20, and Fig.2h**). (please see *Questions #2 and #7* for details)

For the second question, we strongly agree with the reviewer’s suggestions. When the excitation light impinges on the surface of the biological tissues, four processes related to light-tissue interaction cannot be ignored, including interface reflection, in-tissue scattering, in-tissue absorption, and tissue autofluorescence. The results of these processes determine penetration depth (*Nat. Biomed. Eng.* 1, 0010 (2017).). Compared with the visible region (Vis)/first near-infrared (NIR-I) window, NIR-II fluorescence imaging has allowed the investigations of deep anatomical features because of the

reduced photon scattering, low photo-absorption, and low auto-fluorescence of tissue (*Nat. Biomed. Eng.* 1, 0010 (2017)). It is less affected by the different thicknesses and scattering coefficient of biological tissues in NIR-II imaging. Meanwhile, semi-quantitative analysis of biological tissue distribution using fluorescence imaging is also a common method (*Nature communications.* 11,3102 (2020); *Bioconjugate Chem.* 2018, 29, 3833–3840; *Adv. Funct. Mater.* 2020, 30, 1906343; *Angew. Chem. Int. Ed.* 2020, 59, 3691 – 3698; *Adv. Healthcare Mater.* 2020, 1901470; *ACS Appl. Mater. Interfaces* 2020, 12, 20281–20286).To further reduce the interference of these factors (tissue scattering, in-tissue absorption, and tissue autofluorescence), the biological tissues were homogenized, and the results are as follows.

To measure the biodistribution of FT-TQT and FT-TQT@FBS, the vital organs (heart, liver, spleen, lung, kidney) were homogenized and measured under the InGaAs camera. **(Figure S13, S14).**

Continued on Next Page

Figure S13. Toxicity and biodistribution assay. **a)** *Ex vivo* NIR-II imaging of the vital organs 7 and 14 days post-injection. Histogram of the fluorescence intensities of the organs and tissues before **(b)** and after **(c)** homogenizing. Data as mean values \pm s.d. ($n = 3$). **d)** Representative major organ histology (H&E stained) of FT-TQT treated mice at 7 and 14 days postinjection. Scale bar: 100 μ m.

Figure S14. Toxicity and biodistribution assay of FT-TQT@FBS. **a)** *Ex vivo* NIR-II imaging of the vital organs 7 and 14 days post-injection. Histogram of the fluorescence intensities of the organs and tissues before **(b)** and after **(c)** homogenizing. Data as mean values \pm s.d. (n = 3). **d)** Representative major organ histology (H&E stained) of FT-TQT@FBS treated mice at 7 and 14 days post-injection. Scale bar: 100 μ m. (Laser, 808 nm; Power, 90 mW \cdot cm⁻²; Exposure time, 30 ms; 1000 nm Long Pass filter).

To make it clear to the reviewer and the readers, we have added the following contents on **Page 14** in the revised manuscript.

To evaluate the biocompatibility of FT-TQT@FBS, NIR-II imaging of the vital organs was performed after intravenous administration at 7 and 14 days (Figure S14).

Compared with FT-TQT, FT-TQT@FBS is most widely distributed in the liver, which has a large particle size and is hard to excrete through the kidney. However, FT-TQT is mainly distributed in the kidney, followed by the liver. Additionally, the H&E staining of the major organs indicated no noticeable pathological change after FT-TQT@FBS treatment.

11. Table S1, does the ϵ value was tested in MeOH or H₂O? Please provide ϵ and brightness values both in MeOH and H₂O

Response: Thanks for the reviewer's valuable comments. In previous Table S1, the ϵ value was tested in H₂O. Here, we have tested the ϵ value in MeOH. The results are as follows.

Table S1. Quantum yield and brightness values of FT-BBT, FT-TQT and FT-TQT@FBS complexes in MeOH and H₂O.

Fluorophore	QY _{MeOH}	QY _{H2O}	ϵ_{H2O}	ϵ_{MeOH}	$\epsilon^*\Phi_{H2O}$	$\epsilon^*\Phi_{MeOH}$
	(%)	(%)	($\times 10^3 M^{-1} \cdot cm^{-1}$)	($\times 10^3 M^{-1} \cdot cm^{-1}$)	($M^{-1} \cdot cm^{-1}$)	($M^{-1} \cdot cm^{-1}$)
FT-BBT	0.23	---	11.4	16.25	---	37.4.
FT-TQT	0.49	0.025	8.94	10.66	2.24	52.2
CH-4T	0.11	0.06	3.03	9.88	1.82	10.87
FT-TQT@FBS	---	0.20	---	---	---	---
IR-26(DCE)	0.05					

REVIEWERS' COMMENTS

Reviewer #1 (Remarks to the Author):

The authors made their effort to properly address my questions and it is ready to be published.

Reviewer #2 (Remarks to the Author):

All concerns have been addressed.

REVIEWERS' COMMENTS

Reviewer #1 (Remarks to the Author):

The authors made their effort to properly address my questions and it is ready to be published.

Response: We appreciate the reviewer very much for your valuable comments and suggestions.

Reviewer #2 (Remarks to the Author):

All concerns have been addressed.

Response: Thanks for your comment. We would like to appreciate the reviewer for your valuable suggestions on our manuscript.